# Analysis of the Agricultural Economy and Agricultural Pollution Using the Decoupling Index in Chengdu, China

**DOI:** 10.3390/ijerph16214233

**Published:** 2019-10-31

**Authors:** Sipan Li, Qunxi Gong, Shaolei Yang

**Affiliations:** 1Business School, Sichuan University, Chengdu 610064, China; lisipan_scu@163.com (S.L.);; 2Chinese Studies Center, Sichuan University, Chengdu 610065, China

**Keywords:** agricultural economy, agricultural pollution, decoupling index, China

## Abstract

The development of the agricultural economy has brought the agricultural pollution, and agricultural pollution has restrained the growth of the agricultural economy and endangered public health. Since the Economic Reform opened up, China has promulgated many policies that have benefited farmers, developed the agricultural economy, and improved agricultural material levels. However, the growth in the agricultural economy has also resulted in significant environmental pollution from pesticides, agricultural fertilizers and agricultural plastic film. Agricultural pollution can affect public health, therefore it is necessary to study the relationship between agricultural economy and agricultural pollution. In this paper, the relationship between the agricultural economy and agricultural pollution is studied by using decoupling index method, taking Chengdu, Sichuan Province, China as an example, and verified by the data of 2008–2017 years of Chengdu. It was found that the decoupling index for pesticide use and per capita gross agricultural production showed “strong decoupling” ideal states in six years. From 2009–2017, the value of it increased from 0.065743–1.597385, and its decoupling state gradually transited to stable strong decoupling. The chemical fertilizer use was found to have a “strong decoupling” ideal state in five years. The value of it decreased from 1.028854–(−0.512702), while its state gradually changed to strong decoupling. However, agricultural plastic film only has one “strong decoupling” state with its value of decoupling index changed from −0.380824–(−0.250250). The increased use of plastic film has resulted in increased agricultural pollution. The research results show that growing with the development of Chengdu’s agricultural economy, the use of pesticides and fertilizers has been decreasing. Chengdu’s agricultural economy and agricultural pollution were found to have a healthy and coordinated development. The development of agricultural economy has not caused much damage to the agricultural environment. In addition, with the decline of agricultural pollution, public health problems will also be reduced.

## 1. Introduction

Agriculture is the foundation of the Chinese national economy as it provides materials for both residents and other industry sectors. As China is a large agricultural country, agriculture is vital to both the Chinese government and the people [1]. Since the Economic Reform and open up in 1978, China’s agricultural economy has achieved great success with most problems that had been associated with insufficient agricultural product supply being solved. In 2017, China’s total grain output reached 617.91 million tons [2], twice the total grain output of 304.765 million tons in 1978. However, the measures taken to increase agricultural production have resulted in the excessive use of chemical fertilizers and pesticides can no longer meet the requirements of the development of agricultural economy in China under the new situation [3]. Due to the increase in agricultural pollution, such as chemical fertilizer pollution, pesticide pollution and intensive farm pollution, which has resulted in the bad situation that nitrogen, phosphorus, pesticides and other organic or inorganic pollutant entering the surface water, groundwater, and soil environments through surface runoff [4]. All these pollutions endanger the environment and the health of residents [5]. Modern agriculture depends heavily on pesticide use and chemical fertilizer, which have successfully increased productivity but led to increasing concerns regarding the environment and human health [6]. The promotion of agricultural economy may bring agricultural pollution, and agricultural pollution may restrain the development of agricultural economy. The relationship between them has an impact on the development of agriculture and public health. Therefore, it is necessary to study the relationship between economic growth and environmental pollution as the destruction of the ecological environment can restrict economic development. Chengdu is a core city in the southwest of China. It has a good level of agricultural development and a high agricultural status in the western region. However, there is no analysis of the relationship between the agricultural economy and the agricultural environment in Chengdu in the previous literature. In order to better measure the relationship between agricultural economy and agricultural pollution, realize the sustainable development of the agricultural economy and explore public health issues, this paper use decoupling index [7] to examine this relationships and provide a new model for the study of economic growth and environmental pollution.

Most research on the decoupling index has involved studying the relationships between economic growth, carbon emissions, energy consumption and other factors. Qiu [8] believes that promoting the coordinated development of economy and environment is the core of urban sustainable development, so he decoupled the economic growth and resource environment of coal cities. Zhang [9,10] and Zhou [11] believed that China has paid a lot of environmental costs in the past development process. Energy conservation and emission reduction are the top priority. They all used decoupling indices to study economic growth and resource consumption and economic growth and energy consumption. Yi [12] studied tourism economic growth and ecological environmental pressure issues in geoparks. After studying the economic growth and building land Liu [13] believed that the dependence of economic growth on construction land resources is gradually decreasing, and manufacturing industry is an important industry to stimulate China’s economy. The analysis of manufacturing economic growth and energy consumption [14] shows that the development effect of low-carbon industry produced by technological progress is the best. The decoupling index is also used in the analysis of economic growth and sustainability. Through Zhang’s analysis [15], it is concluded that the sustainability of China’s economic growth increases with economic growth. Long [16] refers to resource consumption and environmental pollution as ecological loss and analyzed the economic growth, ecological loss and welfare levels in the process of urbanization. The remaining scholars use the decoupling index to analyze the relationship between economy and environment, such as the impact of economic, energy and environmental factors on eco-efficiency performance indicators [17], economic growth and groundwater consumption [18], economic growth and environmental pollution [19], economic growth and environmental pressure [20], and economic growth and environmental conditions [21].

Other research on carbon emissions or energy consumption have examined the relationships between industrial growth and carbon emissions [22], industrial development and energy consumption, carbon dioxide emissions [23], regional high-tech industry agglomeration and the ecological environment [24]. These articles are based on the impact of the development of an industry on carbon emissions. Some scholars look for the relationship between carbon emissions and urban development. Xiang [25] studied the urbanization and the associated carbon intensity and concluded that urbanization was rising and carbon emissions were declining. From the environmental point of view are environmental regulations and carbon emissions [26] and regional welfare levels and natural consumption [27]. The remaining scholars studied it from the perspective of economic development or transportation, including economic growth and traffic carbon dioxide emissions [28], industrial carbon emissions and economic growth [29], carbon dioxide emissions and total electricity generation [30], transportation carbon dioxide emissions and economic growth [31], and carbon dioxide emissions and economic growth in the maritime industry [32].

In summary, the decoupling index has been widely used to examine economic and ecological relationships in industry, tourism, manufacturing, high-tech industries and other fields. However, that has been little research that has used the decoupling index in agricultural relationship studies even though it is suitable for examining the relationships between agricultural economic growth and agricultural pollution. Past studies have generally chosen carbon dioxide emissions as the measurement criterion for environmental pollution, but have failed to consider pollution-related agricultural indicators, such as pesticide and agricultural fertilizer use.

Therefore, from the perspective of agriculture, this paper obtains the data of agricultural economy and agricultural pollution in Chengdu Statistical Yearbook [33], and puts forward a new evaluation system to evaluate the relationship between agricultural economy and agricultural pollution with the method of decoupling index.

## 2. Materials and Methods

### 2.1. Decoupling Exponential Model

The “decoupling” concept originated from physics and was defined as the absence of relationships between two or more physical quantities based on response relationships. The decoupling index has usually been used to measure the relationships and non-synchronous changes between resource consumption, environmental pressure, and economic growth [16]. For example, if the economic wealth growth rate in a certain period is faster than the environmental degradation rate caused by economic activities, there is a decoupling relationship between them [12]. The Organization for Economic Cooperation and Development (OECD) has generally used the decoupling index in its agricultural policy research [34], and has divided the “decoupling” into absolute decoupling and relative decoupling. Absolute decoupling, also known as strong decoupling, refers to the stable or declining state of the environmental variables associated with economic development. Relative decoupling, also known as weak decoupling, refers to the increase in environmental pressure and resource consumption and economic development, with the extent being less than the extent of economic growth [23]. In 2018, due to the long-term rapid economic and social development, the polluting of the agricultural ecological environment was found to be serious. Therefore, to protect the ecological environment and ensure the quality and safety of agricultural products, Sichuan Province issued the Regulations on the Protection of Agricultural Ecological Environment to establish a supervision and management mechanism for agricultural ecological environment protection, a dynamic monitoring and evaluation system for agricultural environmental protection, a pollution prevention and control planning system, and a classification and protection system for cultivated land.

Therefore, to explore the specific relationships between Chengdu’s economic development and its environmental protection, based on data from the Chengdu Statistical Yearbook [33] from 2008–2017, this paper selected the per capita gross agricultural production value, and pesticide, agricultural fertilizer, and agricultural plastic use to calculate the decoupling index.

### 2.2. Index Selection Logic

Per capita gross agricultural production was selected to reflect the economic growth in agriculture as it can reveal the total scale of regional agricultural production and economic growth. The indicators selected to reveal the agricultural pollution problems were the use of pesticides, agricultural fertilizers and agricultural plastic film. China is the world’s largest pesticide user, with the pesticide use area being 2.5 times the world average [5]. China uses 40% of the world’s chemical fertilizers; however, as the organophosphorus, organochlorines, heavy metals and other toxins from pesticides, herbicides and some fertilizers can pollute water bodies [4], pesticide pollution is a significant agricultural pollution source and its index has important reference value. The excessive use of chemical fertilizers can also seriously pollute surface and groundwater. Therefore, chemical fertilizers are another major source of agricultural pollution. The extensive use of agricultural film in agricultural production has been found to accelerate the “death” of cultivated land, and its degradation releases carcinogenic dioxins into the air, endangering life and health [5]. In the past literature [35,36], the above indicators are also used to measure agricultural environmental pollution. Therefore, this paper used agricultural plastic film as another index to evaluate agricultural pollution.

### 2.3. Measurement Method and Steps in the Decoupling Index

Taking the OECD decoupling index [34] as a reference, a decoupling index measurement model was constructed to assess the agricultural economy and the associated agricultural pollution, the specific calculation steps and technical roadmap for which are shown in Figure 1.

#### 2.3.1. The Calculating Process and Formula of Decoupling Index

The formula for calculating the decoupling index is as follows:(1)$$T_{i}=\frac{\Delta P_{i}}{P_{i}}/\frac{\Delta G}{G}$$
where: $T_{i}$ indicates the decoupling Index for the agricultural economy and agricultural pollution. G and $P_{i}$ indicate the per capita gross agricultural production value and agricultural pollution indicators respectively. ΔG and ΔP indicate the change value for the per capita gross agricultural production value and agricultural pollution indicators respectively. $\frac{\Delta G}{G}$ and $\frac{\Delta P_{i}}{P_{i}}$ indicate the change rate of per capita gross agricultural production value and the change rate of agricultural pollution indicators respectively.

The value of i is 1, 2 or 3, when i = 1, $T_{1}$ indicates the decoupling index for pesticide use and per capita gross agricultural production, $P_{1}$ indicates the pesticide use; when i = 2, $T_{2}$ indicates the decoupling index for the amount of fertilizer used in agriculture and per capita gross agricultural production, $P_{2}$ indicates the agricultural fertilizer application; when i = 3, $T_{3}$ indicates the decoupling index for agricultural plastic film use and per capita gross agricultural production value, $P_{3}$ indicates the agricultural plastic film usage.

#### 2.3.2. Calculating the Variation Values for the Indicators

Based on the selected indicators, the change values in each index in the raw data in each year were calculated and compared with the previous year. The change values for the raw data and each index and the comparisons with the previous year are shown in Table 1 and Table 2.

Second, the annual change rate for each index was calculated based on the values in Table 2. per capita gross agricultural production value, pesticide use, agricultural fertilizer use and agricultural plastic film use. The calculated rate of change is shown in Table 3.

#### 2.3.3. Calculating the Decoupling Index for the Agricultural Economy and Agricultural Pollution

Finally, the decoupling indexes for pesticide use and per capita gross agricultural production, agricultural fertilizer use and per capita gross agricultural production, and agricultural plastic film use and per capita gross agricultural production were calculated using the decoupling index formula based on the calculated rate of change.

The calculated decoupling index is shown in Table 4.

## 3. The Decoupling Index Calculation Results

### 3.1. Decoupling Index State Classification

Depending on the differences in the elastic coefficient, the Tapio tool [7] divides the decoupling state into eight types: strong, strong negative, weak decoupling, weak negative, expansive negative, expansive connection, recessive decoupling and recessive connection [12]. The specific evaluation criteria are shown in Table 5.

### 3.2. Calculation Results

Based on the Tapio decoupling index classification [7] in Table 5, the decoupling index calculations in Table 4 were corresponded one by one, and the specific classifications for each decoupling index in each year determined, as shown in Table 6.

Strong decoupling indicates that the growth rate of the former is negative and that of the latter is positive, which is the ideal state for decoupling development. Weak decoupling indicates that the growth rate of the former is less than that of the latter. The worst decoupling state is an expansive negative decoupling, which indicates that both the former and latter are growing at the same time, but the former is growing faster than the latter. Expansive negative decoupling indicates typical extensive growth [16].

The decoupling index for pesticide use and per capita gross agricultural production, decoupling index for the amount of fertilizer used in agriculture and per capita gross agricultural production and decoupling index for agricultural plastic film use and per capita gross agricultural production are shown in Figure 2, Figure 3 and Figure 4 respectively.

## 4. Discussion

### 4.1. Results Discussion

Pesticide use contributes to increased agricultural production and improved product quality, but overuse or abuse of pesticides has been related to negative externalities such as human health effects and external ecological effects [6]. Heavy metals may be added to soils in agricultural fertilizers and pesticides, soil amendments, or organic fertilizers [37].

The decoupling index for Chengdu from 2009–2012 was found to be in a stable state in terms of pesticide use and per capita gross agricultural production values there was a “strong decoupling” state for three years, which indicated that during this period, the use of pesticides was declining with the economic growth; that is, the increase in the agricultural economy during this period did not result in much agricultural pollution. In 2013, although the use of pesticides was still declining, there was a “declining decoupling” state due to the negative growth in per capita agricultural production in Chengdu in this year. After 2013, the per capita gross agricultural production in Chengdu was annually increasing. The use of pesticides rebounded in 2016, but for the rest of the year was still decreasing; therefore, the use of pesticides and the per capita gross agricultural production values in 2014, 2015 and 2017 had good “strong decoupling” and only one “expansive negative decoupling” in 2016.

The fertilizer application and per capita gross agricultural production decoupling index was fluctuating with no obvious upward or downward trends. From 2009–2012, as the agricultural chemical fertilizer use declined gradually and the per capita gross agricultural production rose, the decoupling index had a short downward trend. In the last four years, two years had an ideal “strong decoupling” state. From 2012–2015, between 15 and 16 t of agricultural chemical fertilizer was used and the annual decline was not significant. The per capita gross agricultural production value began to rise after the decline in 2013, resulting in  an  ideal “strong decoupling” state in 2014 and 2015. In 2016, the agricultural chemical fertilizer use increased sharply to more than 18 tons, which was significantly larger than the growth in the per capita gross agricultural production value; therefore, there was “expansive negative decoupling”.

The agricultural plastic film use and per capita gross agricultural production were also found to have a fluctuating trend, most of which indicated that there was a “weak decoupling” situation, with only one year indicating a “strong decoupling” ideal state, and the remaining four years being unsatisfactory. This was because except for 2017, agricultural plastic film use was increasing. In 2010 and 2014, the growth in agricultural plastic film use exceeded the growth in per capita gross agricultural production; therefore, overall, there were two years of “expansive negative decoupling” with the other years showing “weak decoupling” states.

### 4.2. Results Analysis

The pesticide use and per capita gross agricultural production decoupling index was relatively optimistic, with six “strong decoupling” years as the pesticide use was decreasing and the per capita gross agricultural production was increasing in most years, which indirectly indicated that the growth in the agricultural economy had little effect on agricultural pollution. The fertilizer use rate had a parabolic change from 2008 to 2017, a sharp rise (2016), and a sharp drop (2012), which resulted in an overall fluctuation in the decoupling index. However, because of the annual increases in the economy, the chemical fertilizer use had a short period of “strong decoupling” after each sharp rise or fall. The increasing use of agricultural plastic film has aggravated agricultural pollution. However, due to the reduction of the other two indicators, the relationship between agricultural pollution and agricultural economy shows a healthy and coordinated state. Agricultural pollution has not increased as a result of the growth of the agricultural economy.

## 5. Conclusions

This paper is devoted to exploring the relationship between agricultural economic development and agricultural pollution, and discussing whether it endangers public health. Therefore, the main innovations of this paper are as follows. The relationship between economic growth and environmental pollution was studied from an agricultural perspective in Chengdu, a core agricultural city in Western China, using a decoupling index to explore the key factors affecting the decoupling effect and to identify the direct relationships between the agricultural economy and agricultural pollution. A new model for measuring agricultural economy and agricultural pollution is proposed and validated with the data from 2008–2017. The following conclusions were drawn from the analyses.

The agricultural economy in Chengdu, Sichuan Province, China was found to be developing each year and the agricultural pollution indicators had variable changes. The use of pesticides has been declining for almost a decade. The decoupling index for pesticide use and per capita gross agricultural production showed ideal state of “strong decoupling” in six years. From 2009–2017, the decoupling index of pesticide use and per capita gross agricultural production increased from 0.065743–1.597385, and its decoupling state gradually transited from weak negative decoupling to stable strong decoupling. The decoupling index for agricultural chemical fertilizer use and per capita agricultural production which showed “strong decoupling” in five years was also in a relatively ideal state. The value of it decreased from 1.028854–(−0.512702), while its state changed to strong decoupling. Although the chemical fertilizer use showed a parabolic changing state over the ten years, after each sharp rise and fall, it was in a gentle decline, which meant that there were “strong decoupling” states in some years. However, in recent years, the use of plastic film has shown a bad trend with its value of decoupling index changed from −0.380824–(−0.250250). Because of the increase in plastic film use, there were some “strong negative decoupling” years and some “expansive negative decoupling” years; however, around 50% of the time, the growth rate in plastic film use did not exceed the growth rate in the per capita gross agricultural production, which was why there were “weak decoupling” states for half the time. Generally speaking, the agricultural pollution in Chengdu was declining slowly, which coupled with the annual increase in the agricultural economy, indicated generally healthy and ideal trends. Therefore, the factor changes indicated that the main pollution indicators and the agricultural economic indicators were showing a relatively ideal trend. This shows that the use of pesticides and fertilizers in agricultural development in Chengdu has been well controlled. Although agricultural plastic film usage had an upward trend, the agricultural economic indicator growth was faster than the agricultural pollution indicator growth, which was an ideal decoupling index relationship. The use of pesticides and fertilizers has increased agricultural productivity, while it has a negative impact on public health. Pesticides and fertilizers can cause water, soil and air pollution, add heavy metals to the soil and may further cause food security problems. The situation of agricultural pollution in Chengdu has not worsened with the development of agricultural economy. Effective control of agricultural pollution in Chengdu will have a positive impact on public health.

The relationship between the agricultural economic development and public health needs to be explored to determine the impact of the changes in the regional agricultural economy and agricultural pollution on resident health and food security. The influencing factors for the changes in the agricultural pollution indicators also need to be further explored.

## Figures and Tables

**Figure 1 ijerph-16-04233-f001:**
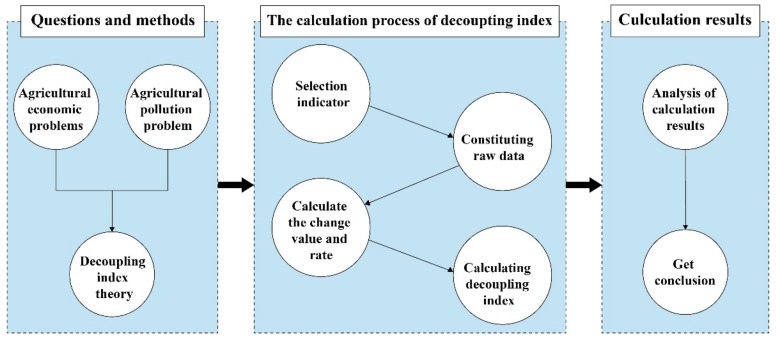
Research framework for decoupling analysis of agricultural economy and agricultural pollution.

**Figure 2 ijerph-16-04233-f002:**
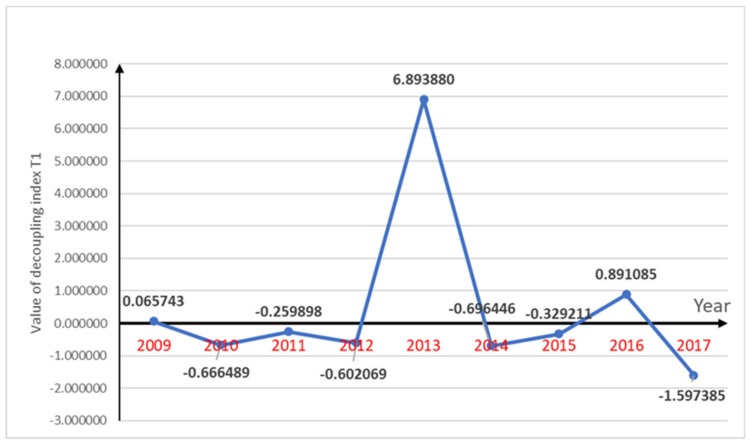
Decoupling index changes of $T_{1}$ along with period of year.

**Figure 3 ijerph-16-04233-f003:**
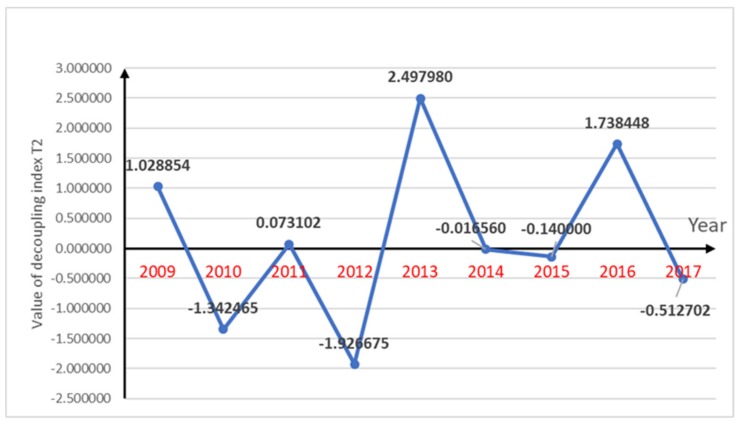
Decoupling index changes of $T_{2}$ along with period of year.

**Figure 4 ijerph-16-04233-f004:**
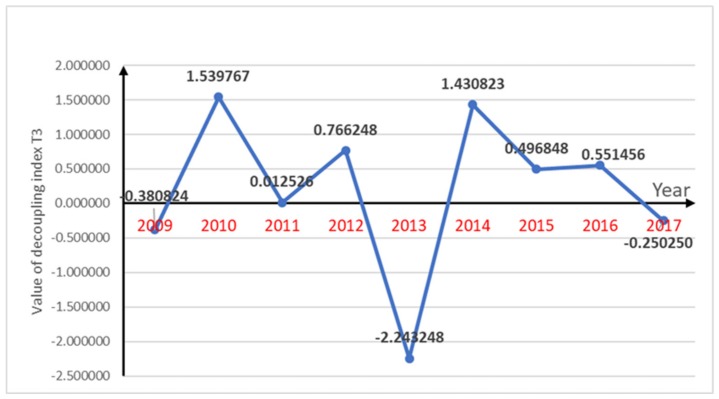
Decoupling index changes of $T_{3}$ along with period of year.

**Table 1 ijerph-16-04233-t001:** Raw data for the selected indicators.

Year	Per Capita Gross Agricultural Production Value/Yuan	Pesticide Use/(Tons)	Agricultural Fertilizer Application/(10,000 Tons)	Agricultural Plastic Film Usage/(Tons)
2008	4153	6145	19.79	9505
2009	3896	6120	18.53	9729
2010	4109	5897	17.17	10,548
2011	4731	5665	17.36	10,568
2012	4946	5510	15.84	10,936
2013	4921	5318	15.64	11,060
2014	5111	5175	15.63	11,671
2015	5438	5066	15.49	12,042
2016	6062	5584	18.58	12,804
2017	6202	5378	18.36	12,730

**Table 2 ijerph-16-04233-t002:** Changed value for the raw data for the selected indicators in each year compared with the previous year.

Year	Per Capita Gross Agricultural Production Value/Yuan	Pesticide Use/(Tons)	Agricultural Fertilizer Application/(10,000 Tons)	Agricultural Plastic Film Usage/(Tons)
2009	−257	−25	−1.26	224
2010	213	−223	−1.36	819
2011	622	−232	0.19	20
2012	215	−155	−1.52	368
2013	−25	−192	−0.2	124
2014	190	−143	−0.01	611
2015	327	−109	−0.14	371
2016	624	518	3.09	762
2017	140	−206	−0.22	−74

**Table 3 ijerph-16-04233-t003:** Rate of change in each index value in each year.

Year	$\frac{\Delta G}{G}$	$\frac{\Delta P_{1}}{P_{1}}$	$\frac{\Delta P_{2}}{P_{2}}$	$\frac{\Delta P_{3}}{P_{3}}$
2009	−0.061883	−0.004068	−0.063669	0.023567
2010	0.054671	−0.036438	−0.073394	0.084181
2011	0.151375	−0.039342	0.011066	0.001896
2012	0.045445	−0.027361	−0.087558	0.034822
2013	−0.005055	−0.034846	−0.012626	0.011339
2014	0.038610	−0.026890	−0.000639	0.055244
2015	0.063980	−0.021063	−0.008957	0.031788
2016	0.114748	0.102250	0.199484	0.063279
2017	0.023095	−0.036891	−0.011841	−0.005779

**Table 4 ijerph-16-04233-t004:** Calculated decoupling index.

Year	2009	2010	2011	2012	2013	2014	2015	2016	2017
$T_{1}$	0.065743	−0.666489	−0.259898	−0.602069	6.893880	−0.696446	−0.329211	0.891085	−1.597385
$T_{2}$	1.028854	−1.342465	0.073102	−1.926675	2.497980	−0.016560	−0.140000	1.738448	−0.512702
$T_{3}$	−0.380824	1.539767	0.012526	0.766248	−2.243248	1.430823	0.496848	0.551456	−0.250250

Note: $T_{1}$ indicates the decoupling index for pesticide use and per capita gross agricultural production. $\;T_{2}$ indicates the decoupling index for the amount of fertilizer used in agriculture and per capita gross agricultural production. $T_{3}$ indicates the decoupling index for agricultural plastic film use and per capita gross agricultural production value.

**Table 5 ijerph-16-04233-t005:** Classification of decoupling index.

Status		Index		
		$\frac{\Delta P_{i}}{P_{i}}$	$\frac{\Delta G}{G}$	$\mathrm{Decoupling}\;\mathrm{Index}\;T_{i}$
Negative Decoupling	expansive	>0	>0	$T_{i}$ > 1.2
	strong	>0	<0	$T_{i}$ < 0
	weak	<0	<0	0 ≤ $T_{i}$ < 0.8
Decoupling	weak	>0	>0	0 ≤ $T_{i}$ < 0.8
	strong	<0	>0	$T_{i}$ < 0
	recessive	<0	<0	$T_{i}$ > 1.2
Connection	expansive	>0	>0	0.8 ≤ $T_{i}$ ≤1.2
	recessive	<0	<0	0.8 ≤ $T_{i}$ ≤1.2

**Table 6 ijerph-16-04233-t006:** Status of the decoupling index for each indicator.

Year	$T_{1}$	$T_{2}$	$T_{3}$
2009	weak negative decoupling	recessive connection	strong negative decoupling
2010	strong decoupling	strong decoupling	expansive negative decoupling
2011	strong decoupling	weak decoupling	weak decoupling
2012	strong decoupling	strong decoupling	weak decoupling
2013	recessive decoupling	recessive decoupling	strong negative decoupling
2014	strong decoupling	strong decoupling	expansive negative decoupling
2015	strong decoupling	strong decoupling	weak decoupling
2016	expansive connection	expansive negative decoupling	weak decoupling
2017	strong decoupling	strong decoupling	strong decoupling

Note: $T_{1}$ indicates the decoupling index for pesticide use and per capita gross agricultural production. $\;T_{2}$ indicates the decoupling index for the amount of fertilizer used in agriculture and per capita gross agricultural production. $T_{3}$ indicates the decoupling index for agricultural plastic film use and per capita gross agricultural production value.

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
