# Peer review of "Analysis of the Agricultural Economy and Agricultural Pollution Using the Decoupling Index in Chengdu, China"

_ijerph, 2019, doi:10.3390/ijerph16214233_

Round 1

Reviewer 1 Report

It is a resubmitted manuscript so I immediately go to the point. In general, the manuscript is now sufficiently valid. My perplexity is still the language. I believe it could be improved if it were subjected to the revision of a native speaker.

The title now is ok.

The abstract has improved.

Here are some suggestions.

You should add China in the keywords.

Lines 36-37: "The agricultural economy is made up of agricultural economic relationships and agricultural economic activities." Please rephrase or delete this sentence.

Line 37: It would be better to include the year of the reform.

Line 48: "roductivity" add the letter "p".

Lines 43-47: This period does not convince me yet. Furthermore, semicolons are incorrect.

Lines 61-62: "With the further deepening of the global resource" (???) This makes no sense to me.

Consider to replace the term "ecological pollution" with "environmental pollution" throughout the manuscript.

Lines 103-113: [In this paper...] This whole part could be better rewritten.

Table 5. "Negative Decoupling": remove the word "negative" from the status column. It is superfluous!

Table 6. Here, instead, you should write the state in full.

Author Response

Manuscript ID: ijerph-624127

Title: Analysis of the Agricultural Economy and Agricultural Pollution in Chengdu, China

Dear Editors and Reviewers: 

Thank you for your letter and for the reviewers’ comments concerning our manuscript. Those comments are all valuable and very helpful for revising and improving our paper, as well as the important guiding significance to our research. We have studied comments carefully and have made correction which we hope meet with approval.

The main corrections in the paper and the responds to the reviewer’s comments are marked in red and italic as following:

You should add China in the keywords.

Modification explanation: According to your requirements, we have added "China" as the keyword.

Keywords: agricultural economy; agricultural pollution; decoupling index; China (line 30 under revised manuscript).

Lines 36-37: "The agricultural economy is made up of agricultural economic relationships and agricultural economic activities." Please rephrase or delete this sentence.

Modification explanation: After consideration, we decided to delete the sentence.

Line 37: It would be better to include the year of the reform.

Modification explanation: We have added the year to the sentence.

Since the Economic Reform and open up in 1978, China's agricultural economy has achieved great success with most problems that had been associated with insufficient agricultural product supply being solved. (line 35 under revised manuscript).

Line 48: "roductivity" add the letter "p".

Modification explanation: We apologize for our carelessness. The word has been corrected.

Modern agriculture depends heavily on pesticide use and chemical fertilizer, which have successfully increased productivity but led to increasing concerns regarding the environment and human health [6]. (line 47 under revised manuscript)

Lines 43-47: This period does not convince me yet. Furthermore, semicolons are incorrect.

Modification explanation: We have adjusted the sentence and now it's easier to understand.

Because of the increase in agricultural pollution, such as chemical fertilizer pollution, pesticide pollution and intensive farm pollution, which has resulted in the bad situation that nitrogen, phosphorus, pesticides and other organic or inorganic pollutant entering the surface water, groundwater, and soil environments through surface runoff [4]. All these pollutions endanger the environment and the health of residents [5]. Modern agriculture depends heavily on pesticide use and chemical fertilizer, which have successfully increased productivity but led to increasing concerns regarding the environment and human health [6]. (line 41-48 under revised manuscript)

Lines 61-62: "With the further deepening of the global resource" (???) This makes no sense to me.

Modification explanation: After consideration, we deleted this sentence.

Consider to replace the term "ecological pollution" with "environmental pollution" throughout the manuscript.

Modification explanation: According to your opinion, we have replaced all "ecological pollution" with "environmental pollution".

Past studies have generally chosen carbon dioxide emissions as the measurement criterion for environmental pollution, (line 100 under revised manuscript)

The relationship between economic growth and environmental pollution was studied from an agricultural perspective in Chengdu, (line 274 under revised manuscript)

Lines 103-113: [In this paper...] This whole part could be better rewritten.

Modification explanation: We rewrote this part and deleted some unnecessary content. Now the expression of this part is clearer.

Therefore, from the perspective of agriculture, this paper obtains the data of agricultural economy and agricultural pollution in Chengdu Statistical Yearbook [33], and puts forward a new evaluation system to evaluate the relationship between agricultural economy and agricultural pollution with the method of decoupling index. (line 102-105 under revised manuscript)

Table 5. "Negative Decoupling": remove the word "negative" from the status column. It is superfluous!

Table 6. Here, instead, you should write the state in full.

Modification explanation: According to your suggestion, we remove the "negative" in the status column of table 5 and complete the status in Table 6.

Table 5. Classification of decoupling index. (line 197 under revised manuscript)

Table 6. Status of the decoupling index for each indicator. (line 201 under revised manuscript)

Other Improvements:

The Figures have been revised more clearly.

We tried our best to improve the manuscript and made some changes in the manuscript. These changes will not influence the content and framework of the paper.

We appreciate for Editors/Reviewers’ warm work earnestly, and hope that the correction will meet with approval.

Once again, thank you very much for your comments and suggestions.

Sipan Li

Sichuan University

2019-10-20

Reviewer 2 Report

Highlight changes in yellow in a next revision, please. No track changes.

Consider comments in the entire text.

Abstract: as previously stated, I would like to see quantitative data.

Revise international unit system… “304.765 million tonnes”

Again now reference 33:

“I see “Chengdu Statistical Yearbooks” Plural, and then, one reference, and then reference 32 reference, singular… No coherence…

Some of the issues pointed about before continue: example: Figure 2 and several others, absolutely no legend in axis…

Incoherence in the headings continues:

3.2. Results and Discussion

3.3. Result Analysis

?!

(and use results…)

Conclusions need to better translate the findings, namely quantitatively

Although better, to be publishable a text needs to have relevance, authors must further work on comments extensively highlighted until now…

Author Response

Manuscript ID: ijerph-624127

Title: Analysis of the Agricultural Economy and Agricultural Pollution in Chengdu, China

Dear Editors and Reviewers: 

Thank you for your letter and for the reviewers’ comments concerning our manuscript. Those comments are all valuable and very helpful for revising and improving our paper, as well as the important guiding significance to our research. We have studied comments carefully and have made correction which we hope meet with approval.

The main corrections in the paper and the responds to the reviewer’s comments are marked in red and italic as following:

Abstract: as previously stated, I would like to see quantitative data.

Modification explanation: According to your requirements, we have added quantitative data to the abstract.

Abstract: The development of agricultural economy has brought the agricultural pollution, and agricultural pollution has restrained the growth of agricultural economy and endangered public health. Since the Economic Reform and open up, China has promulgated many policies that have benefited farmers, developed the agricultural economy, and improved agricultural material levels. However, the growth in the agricultural economy has also resulted in significant environmental pollution from pesticides, agricultural fertilizers and agricultural plastic film. Agricultural pollution can affect public health, therefore it is necessary to study the relationship between agricultural economy and agricultural pollution. In this paper, the relationship between agricultural economy and agricultural pollution is studied by using decoupling index method, taking Chengdu, Sichuan Province, China as an example, and verified by the data of 2008-2017 years of Chengdu. It was found that the decoupling index for pesticide use and per capita gross agricultural production showed "strong decoupling" ideal states in six years and the chemical fertilizer use was found to have a "strong decoupling" ideal state in five years; however, agricultural plastic film only has one "strong decoupling" state. The increased use of plastic film has resulted in increased agricultural pollution. The research results show that growing with the development of Chengdu's agricultural economy, the use of pesticides and fertilizers has been decreasing. Chengdu's agricultural economy and agricultural pollution were found to have a healthy and coordinated development. The development of agricultural economy has not caused much damage to the agricultural environment. In addition, with the decline of agricultural pollution, public health problems will also be reduced. (line 11-29 under revised manuscript).

Revise international unit system… “304.765 million tonnes”

Modification explanation: We have replaced "tonne" with "ton".

Since the Economic Reform and open up in 1978, China's agricultural economy has achieved great success with most problems that had been associated with insufficient agricultural product supply being solved. In 2017, China's total grain output reached 617.91 million tons [2], twice the total grain output of 304.765 million tons in 1978. (line 35-38 under revised manuscript).

Again now reference 33

Modification explanation: We checked and corrected the mistakes in the singular and plural. Now we use the singular in the text.

Therefore, to explore the specific relationships between Chengdu's economic development and its environmental protection, based on data from the Chengdu Statistical Yearbook [33] (line 128-129 under revised manuscript).

Some of the issues pointed about before continue: example: Figure 2 and several others, absolutely no legend in axis…

Modification explanation: According to your opinion, we have added legend on the axes in figure 2, figure 3 and figure 4.

Figure 2. Annual decoupling index for pesticide use and per capita gross agricultural production (line 221-223 under revised manuscript)

Figure 3. Annual decoupling index for the amount of fertilizer used in agriculture and per capita gross agricultural production (line 235-237 under revised manuscript)

Figure 4. Annual decoupling index for agricultural plastic film use and per capita gross agricultural production (line 248-250 under revised manuscript)

Incoherence in the headings continues: 

3.2. Results and Discussion

3.3. Result Analysis

?!

(and use results…)

Modification explanation: We changed these two headings to "Results discussion" and "Results analysis"

3.2. Results Discussion (line 216 under revised manuscript)

3.3. Results Analysis (line 258 under revised manuscript)

Conclusions need to better translate the findings, namely quantitatively

Modification explanation: According to your suggestion, We added quantitative data to our conclusions.

The agricultural economy in Chengdu, Sichuan Province, China was found to be developing each year and the agricultural pollution indicators had variable changes. The use of pesticides has been declining for almost a decade. The decoupling index for pesticide use and per capita gross agricultural production showed ideal state of "strong decoupling" in six years. The decoupling index for agricultural chemical fertilizer use and per capita agricultural production which showed "strong decoupling" in five years was also in a relatively ideal state. (line 280-285 under revised manuscript)

Other Modifications:

We have adjusted many sentence structures to make the article more fluent.

We tried our best to improve the manuscript and made some changes in the manuscript. These changes will not influence the content and framework of the paper.

We appreciate for Editors/Reviewers’ warm work earnestly, and hope that the correction will meet with approval.

Once again, thank you very much for your comments and suggestions.

Sipan Li

Sichuan University

2019-10-20

Round 2

Reviewer 2 Report

Highlight changes in yellow in a next revision, please. No track changes.

Consider comments in the entire text.

Where is the yellow indicated in the text so that changes can be clearly observed…?

There is really no quantitative data in abstract:

“1. Abstract: as previously stated, I would like to see quantitative data.

Modification explanation: According to your requirements, we have added quantitative data to the abstract.

Abstract: The development of agricultural economy has brought the agricultural pollution, and agricultural pollution has restrained the growth of agricultural economy and endangered public health. Since the Economic Reform and open up, China has promulgated many policies that have benefited farmers, developed the agricultural economy, and improved agricultural material levels. However, the growth in the agricultural economy has also resulted in significant environmental pollution from pesticides, agricultural fertilizers and agricultural plastic film. Agricultural pollution can affect public health, therefore it is necessary to study the relationship between agricultural economy and agricultural pollution. In this paper, the relationship between agricultural economy and agricultural pollution is studied by using decoupling index method, taking Chengdu, Sichuan Province, China as an example, and verified by the data of 2008-2017 years of Chengdu. It was found that the decoupling index for pesticide use and per capita gross agricultural production showed "strong decoupling" ideal states in six years and the chemical fertilizer use was found to have a "strong decoupling" ideal state in five years; however, agricultural plastic film only has one "strong decoupling" state. The increased use of plastic film has resulted in increased agricultural pollution. The research results show that growing with the development of Chengdu's agricultural economy, the use of pesticides and fertilizers has been decreasing. Chengdu's agricultural economy and agricultural pollution were found to have a healthy and coordinated development. The development of agricultural economy has not caused much damage to the agricultural environment. In addition, with the decline of agricultural pollution, public health problems will also be reduced. (line 11-29 under revised manuscript).

I have difficulty in observing the same quantitative data:

“6. Conclusions need to better translate the findings, namely quantitatively

Modification explanation: According to your suggestion, We added quantitative data to our conclusions.

The agricultural economy in Chengdu, Sichuan Province, China was found to be developing each year and the agricultural pollution indicators had variable changes. The use of pesticides has been declining for almost a decade. The decoupling index for pesticide use and per capita gross agricultural production showed ideal state of "strong decoupling" in six years. The decoupling index for agricultural chemical fertilizer use and per capita agricultural production which showed "strong decoupling" in five years was also in a relatively ideal state.

Figures: An xx axis relating years, should have a legend relating “year”…

CNY” should not be used in an international journal…

Previous comments must be considered…

Author Response

Manuscript ID: ijerph-624127

Title: Analysis of the Agricultural Economy and Agricultural Pollution in Chengdu, China

Dear Editors and Reviewers: 

Thank you for your letter and for the reviewers’ comments concerning our manuscript. Those comments are all valuable and very helpful for revising and improving our paper, as well as the important guiding significance to our research. We have studied comments carefully and have made correction which we hope meet with approval.

The main corrections in the paper and the responds to the reviewer’s comments are marked in red and italic as following:

Where is the yellow indicated in the text so that changes can be clearly observed…?

Modification explanation: We indicated the revised parts in the previous manuscript in yellow. The revised parts of our manuscript and reply are all indicated with yellow. We are sorry for the inconvenience to your review.

There is really no quantitative data in abstract:

I have difficulty in observing the same quantitative data:

Modification explanation: We add the analysis of the calculation results to the abstract and conclusion, which shows the calculated quantitative data.

It was found that the decoupling index for pesticide use and per capita gross agricultural production showed "strong decoupling" ideal states in six years. From 2009 to 2017, the value of it increased from 0.065743 to 1.597385, and its decoupling state gradually transited to stable strong decoupling. The chemical fertilizer use was found to have a "strong decoupling" ideal state in five years. The value of it decreased from 1.028854 to -0.512702, while its state gradually changed to strong decoupling. However, agricultural plastic film only has one "strong decoupling" state with its value of decoupling index changed from -0.380824 to -0.250250. The increased use of plastic film has resulted in increased agricultural pollution. The research results show that growing with the development of Chengdu's agricultural economy, the use of pesticides and fertilizers has been decreasing. Chengdu's agricultural economy and agricultural pollution were found to have a healthy and coordinated development. The development of agricultural economy has not caused much damage to the agricultural environment. In addition, with the decline of agricultural pollution, public health problems will also be reduced. (line 22-27 under revised manuscript).

The agricultural economy in Chengdu, Sichuan Province, China was found to be developing each year and the agricultural pollution indicators had variable changes. The use of pesticides has been declining for almost a decade. The decoupling index for pesticide use and per capita gross agricultural production showed ideal state of "strong decoupling" in six years. From 2009 to 2017, the decoupling index of pesticide use and per capita gross agricultural production increased from 0.065743 to 1.597385, and its decoupling state gradually transited from weak negative decoupling to stable strong decoupling. The decoupling index for agricultural chemical fertilizer use and per capita agricultural production which showed "strong decoupling" in five years was also in a relatively ideal state. The value of it decreased from 1.028854 to -0.512702, while its state changed to strong decoupling. Although the chemical fertilizer use showed a parabolic changing state over the ten years, after each sharp rise and fall, it was in a gentle decline, which meant that there were "strong decoupling " states in some years. However, in recent years, the use of plastic film has shown a bad trend with its value of decoupling index changed from -0.380824 to -0.250250. (line 284-296 under revised manuscript).

Figures: An xx axis relating years, should have a legend relating “year”…

Modification explanation: We have changed the font size and color of the year information on the horizontal axis of the Figure 2,3 and 4, making it easier to distinguish from other information. We have modified the name of the Figure 2,3 and 4 to reflect the information in the Figure more accurately.

Figure 2. Decoupling index changes of  along with period of year (line 225-226 under revised manuscript)

Figure 3. Decoupling index changes of  along with period of year (line 238-239 under revised manuscript)

Figure 4. Decoupling index changes of  along with period of year (line 250-251 under revised manuscript)

CNY” should not be used in an international journal…

Modification explanation: According to your opinion, we have replaced all "CNY" with "Yuan".

Other Modifications:

We have adjusted some words.

We tried our best to improve the manuscript and made some changes in the manuscript. These changes will not influence the content and framework of the paper.

We appreciate for Editors/Reviewers’ warm work earnestly, and hope that the correction will meet with approval.

Once again, thank you very much for your comments and suggestions.

Sipan Li

Sichuan University

2019-10-22

This manuscript is a resubmission of an earlier submission. The following is a list of the peer review reports and author responses from that submission.

Round 1

Reviewer 1 Report

The manuscript "A Decoupling Analysis of the Agricultural Economy and Agricultural Pollution in Chengdu, Sichuan Province, China" deals with the important issue of sustainable development through the use of the decoupling index between agricultural economy and agricultural pollution. The theme is not new but the novelty lies in the application of the index to this context.

However, I think the work was presented poorly and hastily and needs a revision of English.

The introduction needs substantial changes and additions.
The literature review presented almost as a list of publications is really bad.
You could also combine sections 1 and 2 and better rework everything.

Line 30: In which year did the reform take place? What changes did you introduce? You should explain this better.

I believe it's more useful to the reader to rename section 3 with "Materials and Methods".
There is no focus on the study area and there are many repetitions.

The results seem quite clear, while in the conclusions I read very few future prospects. I think it is more useful to rename the last section in "Conclusions" or it's necessary to treat the perspectives in a better way.

Author Response

Manuscript ID: ijerph-593479

Title: Analysis of the Agricultural Economy and Agricultural Pollution in Chengdu, Sichuan Province, China

Dear Editors and Reviewers: 

Thank you for your letter and for the reviewers’ comments concerning our manuscript. Those comments are all valuable and very helpful for revising and improving our paper, as well as the important guiding significance to our research. We have studied comments carefully and have made correction which we hope meet with approval.

The main corrections in the paper and the responds to the reviewer’s comments are marked in red and italic as following:

The introduction needs substantial changes and additions. The literature review presented almost as a list of publications is really bad. You could also combine sections 1 and 2 and better rework everything.

Modification explanation: Thank you for careful examination of the article. We have revised and supplemented the introduction. In the introduction, we clarify the research objectives and significance of this paper, and highlight the innovative points of this paper. Following your suggestion, we have merged sections 1 and 2 together and made a lot of adjustments to the literature review section. The research on decoupling index can be divided into two categories: one is about the relationship between economic growth and energy consumption, the other is about the relationship between carbon emissions and the development of an industry. We present the important viewpoints of scholars in the literature review, and re-classify the literature to make the content of the literature review more comprehensive and clear.

As the destruction of the ecological environment can restrict economic development. Therefore, it is necessary to study the relationship between economic growth and environmental pollution. Chengdu is a core city in the southwest of China. It has a good level of agricultural development and a high agricultural status in the western region. However, there is no analysis of the relationship between the agricultural economy and the agricultural environment in Chengdu in the previous literature. In order to better measure the relationship between agricultural economy and agricultural pollution and realize the sustainable development of the agricultural economy, this paper use decoupling index [6] to examine the relationships between the agricultural economy and agricultural pollution and provide a new model for the study of economic growth and environmental pollution. Thus, the main innovations of this paper are as follows. The relationship between economic growth and ecological pollution was studied from an agricultural perspective using a decoupling index to explore the key factors affecting the decoupling effect and to identify the direct relationships between the agricultural economy and agricultural pollution. Agricultural environmental pollution indicators were selected; pesticides, agricultural fertilizers and agricultural plastic film; which better reflected the characteristics of agricultural pollution. 

With the further deepening of the global resource and environmental crisis, economic, energy and environmental issues have become the focus of domestic and foreign scholars. Most research on the decoupling index has involved studying the relationships between economic growth, carbon emissions, energy consumption and other factors. Qiu [7] believes that promoting the coordinated development of economy and environment is the core of urban sustainable development, so he decoupled the economic growth and resource environment of coal cities. Zhang [8, 9] and Zhou [10] believed that China has paid a lot of environmental costs in the past development process. Energy conservation and emission reduction are the top priority. They all used decoupling indices to study economic growth and resource consumption and economic growth and energy consumption. Yi [11] studied tourism economic growth and ecological environmental pressure issues in geoparks. After studying the economic growth and construction land Liu [12] believed that the dependence of economic growth on construction land resources is gradually decreasing, and manufacturing industry is an important industry to stimulate China's economy. The analysis of manufacturing economic growth and energy consumption [13] shows that the development effect of low-carbon industry produced by technological progress is the best. The decoupling index is also used in the analysis of economic growth and sustainability. Through Zhang's analysis [14], it is concluded that the sustainability of China's economic growth increases with economic growth. Long [15] refers to resource consumption and environmental pollution as ecological loss and analyzed the economic growth, ecological loss and welfare levels in the process of urbanization. The remaining scholars use the decoupling index to analyze the relationship between economy and environment, such as the impact of economic, energy and environmental factors on eco-efficiency performance indicators [16], economic growth and groundwater consumption [17], economic growth and environmental pollution [18], economic growth and environmental pressure [19], and economic growth and environmental conditions [20].

Other research on carbon emissions or energy consumption have examined the relationships between industrial growth and carbon emissions [21], industrial development and energy consumption, carbon dioxide emissions [22], regional high-tech industry agglomeration and the ecological environment [23]. These articles are based on the impact of the development of an industry on carbon emissions. Some scholars look for the relationship between carbon emissions and urban development. Xiang [23] studied the urbanization and the associated carbon intensity and concluded that urbanization was rising and carbon emissions were declining. From the environmental point of view are environmental regulations and carbon emissions [25] and regional welfare levels and natural consumption [26]. The remaining scholars studied it from the perspective of economic development or transportation, including economic growth and traffic carbon dioxide emissions [27], industrial carbon emissions and economic growth [28], carbon dioxide emissions and total electricity generation [29], transportation carbon dioxide emissions and economic growth [30], and carbon dioxide emissions and economic growth in the maritime industry [31]. (line 40-92 under revised manuscript).

Line 30: In which year did the reform take place? What changes did you introduce? You should explain this better.

Modification explanation: We have supplemented this part. We added the specific year of China's Economic Reform and open up and the impact of Economic Reform and open up on agriculture, and added references to support it.

Since the Economic Reform and open up, China's agricultural economy has achieved great success with most problems that had been associated with insufficient agricultural product supply being solved. In 2017, China's total grain output reached 617.91 million tonnes [2], twice the total grain output of 304.765 million tonnes in 1978. (line 31-35 under revised manuscript).

I believe it's more useful to the reader to rename section 3 with "Materials and Methods".
There is no focus on the study area and there are many repetitions.

Modification explanation: We have renamed the title with "Materials and Methods" and adjusted the subtitle and the structure of the calculation process for this part.

We have revised the literature review and classified the literature more clearly so that readers can understand the field of decoupling index research more easily. In the part of index selection logic, we trimmed the text content, and added references to support the selection of indicators.

Materials and Methods (line 110 under revised manuscript).

China is the world's largest pesticide user, with the pesticide use area being 2.5 times the world average [5]. China uses 40% of the world's chemical fertilizers; however, as the organophosphorus, organochlorines, heavy metals and other toxins from pesticides, herbicides and some fertilizers can pollute water bodies [4], pesticide pollution is a significant agricultural pollution source and its index has important reference value. The excessive use of chemical fertilizers can also seriously pollute surface and groundwater. Therefore, chemical fertilizers are another major source of agricultural pollution. The extensive use of agricultural film in agricultural production has been found to accelerate the "death" of cultivated land, and its degradation releases carcinogenic dioxins into the air, endangering life and health [5]. In the past literature[34, 35], the above indicators are also used to measure agricultural environmental pollution. (line 143-152 under revised manuscript).

The results seem quite clear, while in the conclusions I read very few future prospects. I think it is more useful to rename the last section in "Conclusions" or it's necessary to treat the perspectives in a better way.

Modification explanation: Thank you for thoughtful consideration. We revised the title of this section and adjusted the conclusion.

Conclusions

This paper used the decoupling index to study the relationship between economic growth and environmental pollution from agricultural perspective in Chengdu, a core agricultural city in Western China. A new model for measuring agricultural economy and agricultural pollution is proposed and validated with the data from 2008 to 2017. The following conclusions were drawn from the analyses. (line 249-254 under revised manuscript)

Other Improvements:

We have supplemented some references to proper nouns. Tables and figures are now are a lot clearer to read and easier to understand

We tried our best to improve the manuscript and made some changes in the manuscript. These changes will not influence the content and framework of the paper.

We appreciate for Editors/Reviewers’ warm work earnestly, and hope that the correction will meet with approval.

Once again, thank you very much for your comments and suggestions.

Sipan Li

Sichuan University

2019-9-20

Reviewer 2 Report

Dear Editor

The paper s written in good English and the topic is interesting and timing, the methodology is appropriate, nevertheless, the paper is very poor. It is possible to argue that is a good exercise, but no more. I suggest to don’t publish.

Author Response

Manuscript ID: ijerph-593479

Title: Analysis of the Agricultural Economy and Agricultural Pollution in Chengdu, Sichuan Province, China

Dear Editors and Reviewers: 

Thank you for your letter and for the reviewers’ comments concerning our manuscript. This paper studies the relationship between economic growth and environmental pollution. We study the relationship between them from the agricultural point of view. The selected research object is Chengdu, the capital city of Sichuan Province, a large agricultural province in Western China. This paper focuses on the sustainable development of agricultural economy and the prevention and control of agricultural pollution. Thus we are looking forward to your valuable suggestions.

We have revised the opinions of other reviewers and sincerely hope to receive your specific comments. We have studied comments carefully and have made correction which we hope meet with approval. The main corrections in the paper and the responds to the reviewer’s comments are marked in red and italic as following:

We revised the abstract and added research data to it to make the results more obvious. We replace the keyword "decoupling index theory" with a more accurate "decoupling index".

It was found that the decoupling index for pesticide use and per capita gross agricultural production showed "strong decoupling" ideal states in six years and the chemical fertilizer use was found to have a "strong decoupling" ideal state in five years; however, in the past four years, agricultural plastic film has shown a state of "weak decoupling", which directly leads to the whole improvement of agricultural pollution. The research results show that Chengdu's agricultural economy and agricultural pollution were found to have a healthy and coordinated development. (line 18-24 under revised manuscript).

Keywords:  agricultural economy; agricultural pollution; decoupling index (line 25 under revised manuscript).

We have revised and supplemented the introduction. In the introduction, we clarify the research objectives and significance of this paper, and highlight the innovative points of this paper. Following your suggestion, we have merged sections 1 and 2 together and made a lot of adjustments to the literature review section. The research on decoupling index can be divided into two categories: one is about the relationship between economic growth and energy consumption, the other is about the relationship between carbon emissions and the development of an industry. We present the important viewpoints of scholars in the literature review, and re-classify the literature to make the content of the literature review more comprehensive and clear.

As the destruction of the ecological environment can restrict economic development. Therefore, it is necessary to study the relationship between economic growth and environmental pollution. Chengdu is a core city in the southwest of China. It has a good level of agricultural development and a high agricultural status in the western region. However, there is no analysis of the relationship between the agricultural economy and the agricultural environment in Chengdu in the previous literature. In order to better measure the relationship between agricultural economy and agricultural pollution and realize the sustainable development of the agricultural economy, this paper use decoupling index [6] to examine the relationships between the agricultural economy and agricultural pollution and provide a new model for the study of economic growth and environmental pollution. Thus, the main innovations of this paper are as follows. The relationship between economic growth and ecological pollution was studied from an agricultural perspective using a decoupling index to explore the key factors affecting the decoupling effect and to identify the direct relationships between the agricultural economy and agricultural pollution. Agricultural environmental pollution indicators were selected; pesticides, agricultural fertilizers and agricultural plastic film; which better reflected the characteristics of agricultural pollution. 

With the further deepening of the global resource and environmental crisis, economic, energy and environmental issues have become the focus of domestic and foreign scholars. Most research on the decoupling index has involved studying the relationships between economic growth, carbon emissions, energy consumption and other factors. Qiu [7] believes that promoting the coordinated development of economy and environment is the core of urban sustainable development, so he decoupled the economic growth and resource environment of coal cities. Zhang [8, 9] and Zhou [10] believed that China has paid a lot of environmental costs in the past development process. Energy conservation and emission reduction are the top priority. They all used decoupling indices to study economic growth and resource consumption and economic growth and energy consumption. Yi [11] studied tourism economic growth and ecological environmental pressure issues in geoparks. After studying the economic growth and construction land Liu [12] believed that the dependence of economic growth on construction land resources is gradually decreasing, and manufacturing industry is an important industry to stimulate China's economy. The analysis of manufacturing economic growth and energy consumption [13] shows that the development effect of low-carbon industry produced by technological progress is the best. The decoupling index is also used in the analysis of economic growth and sustainability. Through Zhang's analysis [14], it is concluded that the sustainability of China's economic growth increases with economic growth. Long [15] refers to resource consumption and environmental pollution as ecological loss and analyzed the economic growth, ecological loss and welfare levels in the process of urbanization. The remaining scholars use the decoupling index to analyze the relationship between economy and environment, such as the impact of economic, energy and environmental factors on eco-efficiency performance indicators [16], economic growth and groundwater consumption [17], economic growth and environmental pollution [18], economic growth and environmental pressure [19], and economic growth and environmental conditions [20].

Other research on carbon emissions or energy consumption have examined the relationships between industrial growth and carbon emissions [21], industrial development and energy consumption, carbon dioxide emissions [22], regional high-tech industry agglomeration and the ecological environment [23]. These articles are based on the impact of the development of an industry on carbon emissions. Some scholars look for the relationship between carbon emissions and urban development. Xiang [23] studied the urbanization and the associated carbon intensity and concluded that urbanization was rising and carbon emissions were declining. From the environmental point of view are environmental regulations and carbon emissions [25] and regional welfare levels and natural consumption [26]. The remaining scholars studied it from the perspective of economic development or transportation, including economic growth and traffic carbon dioxide emissions [27], industrial carbon emissions and economic growth [28], carbon dioxide emissions and total electricity generation [29], transportation carbon dioxide emissions and economic growth [30], and carbon dioxide emissions and economic growth in the maritime industry [31]. (line 40-92 under revised manuscript).

We redrawn the technology roadmap and named Figure 1 "Technical roadmap for decoupling analysis of agricultural economy and agricultural pollution." This makes it easier for readers to understand what Figure 1 is intended to express. We have adjusted some of the text expression inside the figure to make the content of the whole technology roadmap more accurate.

Figure 1. Technical roadmap for decoupling analysis of agricultural economy and agricultural pollution. (line 158-160 under revised manuscript).

At the beginning of the calculation part, we add a formula description of the whole calculation process, explaining in detail the meaning of each symbol or term. In this way, the reader can have a detailed understanding of the calculation process and symbolic terms at the beginning, and will not produce ambiguity in the use of symbols or terms later.

The formula for calculating the decoupling index is as follows:

Where:  indicates the decoupling Index for the agricultural economy and agricultural pollution.  and indicate the per capita gross agricultural production value and agricultural pollution indicators respectively.  and  indicate the change value for the per capita gross agricultural production value and agricultural pollution indicators respectively.  and  indicate the change rate of per capita gross agricultural production value and the change rate of agricultural pollution indicators respectively.

The value of  is 1, 2 or 3, when  = 1 indicates the decoupling index for pesticide use and per capita gross agricultural production indicates the pesticide usewhen  = 2 indicates the decoupling index for the amount of fertilizer used in agriculture and per capita gross agricultural production indicates the agricultural fertilizer applicationwhen  = 3 indicates the decoupling index for agricultural plastic film use and per capita gross agricultural production value indicates the agricultural plastic film usage.  (line 161-175 under revised manuscript).

Based on the description of Question 4, after defining specific symbols and terms, we modify the description of the indicators in this paper to make the calculation process and result analysis more standardized and clear. We replace the indicators in tables with symbols or terms, and describe the indicators in the text with full names.

We sorted out the structure of the calculation part, and checked the subtitles, table names and figure names of the full text. We removed some unnecessary subtitles from the text, and adjusted some of the subtitles, table names and figure names. Now the whole calculation process seems to be more smooth, and the revision of some headings makes it easier for readers to understand the text content.

3.1. Decoupling Index State Classification (line 195 under revised manuscript).

Table 2. Changed value for the original data for the selected indicators in each year compared with the previous year. (line 181-182 under revised manuscript).

Based on the selected indicators, the change values in each index in the original data in each year were calculated and compared with the previous year. (line 177-178 under revised manuscript).

2.3.1. The Calculating Process and Formula of Decoupling Index (line 161 under revised manuscript).

Figure 1. Technical roadmap for decoupling analysis of agricultural economy and agricultural pollution. (line 158-160 under revised manuscript).

We revised the title. We have adjusted the conclusions, highlighted the method, objectives and research significance of this paper once again, including the innovation and advantages of this paper. We generalize more briefly in language, and the content of the conclusion corresponds to the revised abstract. Conclusions

This paper used the decoupling index to study the relationship between economic growth and environmental pollution from agricultural perspective in Chengdu, a core agricultural city in Western China. A new model for measuring agricultural economy and agricultural pollution is proposed and validated with the data from 2008 to 2017. The following conclusions were drawn from the analyses. (line 249-254 under revised manuscript).

We add references to some proper nouns and indicators, and check the format and writing of the full text.

Tapio P. Towards a theory of decoupling: degrees of decoupling in the EU and the case of road traffic in Finland between 1970 and 2001J].Transport Policy200512( 2) : 137 151(line 47 under revised manuscript).

To establish the agricultural economic growth and agricultural pollution research model and select the specific indicators, this paper synthesized previous studies on the relationships between economic growth and the ecological environment or ecological pollution, and selected indicators from 2008 to 2017 from the Chengdu Statistical Yearbooks [32]. (line 102-105 under revised manuscript)

Therefore, to explore the specific relationships between Chengdu's economic development and its environmental protection, based on data from the Chengdu Statistical Yearbooks [32] from 2008 to 2017, (line 132-134 under revised manuscript)

Chengdu Statistical Bureau. Chengdu Statistical Yearbook; China Statistics Press: Beijing, China, 2008–2017. (line 368 under revised manuscript)

The Organization for Economic Cooperation and Development (OECD) has generally used the decoupling index in its agricultural policy research [33], (line 117-119 under revised manuscript).

Taking the OECD decoupling index [33] as a reference, a decoupling index measurement model was constructed to assess the agricultural economy and the associated agricultural pollution, the specific calculation steps and technical roadmap for which are shown in Figure 1.  (line 155-157 under revised manuscript).

OECD. Effects of quantitative constraints on the degree of decoupling of crop support measures [R]. Paris: OECD, 2005. (line 369-370 under revised manuscript).

We have revised the words and language style in some parts of the article.

We have redrawn Figure 2 to show more clearly the annual trend of the decoupling indices of the indicators.

Figure 2. Annual decoupling index diagram (line 212-213 under revised manuscript).

We tried our best to improve the manuscript and made some changes in the manuscript. These changes will not influence the content and framework of the paper.

We appreciate for Editors/Reviewers’ warm work earnestly, and hope that the correction will meet with approval.

Sipan Li

Sichuan University

2019-9-20

Reviewer 3 Report

Highlight changes in yellow in a next revision, please. No track changes.

Consider comments in the entire text.

Title: I believe authors must revise the term “Decoupling”

[See that “decoupling index theory” appears ONCE in the text, how can it be chosen as keyword? Revise keywords…]

Abstract: The abstract needs to be clear in terms of results and findings, namely explicitly clarifying meaning and ranges, namely in terms of index and quantitative data.

Reading the abstract it is not clear what kind if index and how it works…

Add practical implications

Revise non clear or scientific expressions: “Overall,”, “relatively healthy”

Revise keywords…, so they better reflect the manuscript

Introduction:

Revise all spacing… “people[1].”

Where are the references in every simiar case? “this paper used the Tapio decoupling index”

Why “decoupling index” and then  also “"decoupling" index”?

Consider revising the way references are presented in the text, there are times when direct referencing style would also be considered not just at the end… “[]”

References? “Chengdu Statistical Yearbooks.” See all similar cases It mentioned and existing, then cite…

To me, this is not the place to add considerations regarding novelty…

“Therefore, the innovations in this study were as 79 follows.”

And please avoid the “listing”…

Section 3:

References?

“The Organization 95 for Economic Cooperation and Development (OECD) has generally used the decoupling index in its 96 agricultural policy research, and has divided the "decoupling" into absolute decoupling and relative 97 decoupling.”

See similar situations

Revise typos… “healthy..”

Revise italics in references: “average [4].”

Outside introduction and literature sections the use of references must be made with care, usually mentioning authors names.

References? “Taking the OECD decoupling index as a reference,”

Figures: PLEASE use proper caption (self-explanatory ones…): “Figure 1. Technical roadmap” says nothing…

Figure is distorted, check consistency and do not use font bigger than text…

Language inside needs revision

Why referring “original data” in

“3.3.1. Calculating the Variation Values for the Indicators in the Original Data”

[used four times in the text]

To me, the language needs clarification and assertiveness:

“First, based on the selected indexes,”

Why use “CNY” in an international indexed Journal?

Check international unit system: “Use/Ton” and then “Application / Tonnes

Units must be indicated inside ()

Change or changed? “Table 2. Change value”

Caption should consider including time period

Then cannot be a second a new subsection, not me:

“3.3.2. Calculating the Change Rate of Each Index in Each Year 143

Second, the annual”

Revise headings and writing

To me mathematical data must be presented (and removed from table using both definition and formula at the same time…) otherwise, as equations (and numbered, au usually…), and terms (parameters) defined and units presented, where available

There are ways to integrate in a vertical table… “Table 4. Calculated decoupling index.”, just further work it…

And more…

The use of so many headings to breaks the text…

And I do feel the language must be clarified:

4.1. Annual Decoupling Index State Classification for Each Index

In tables, use just the nomenclature and notes can give additional information, including the identification... “Decoupling Index ?”

No formatted (justified…) text here: “Based on the TAPIO decoupling”

[and why previously lower letter, still no references… “Tapio decoupling”]

Figure 2: No xx axis label…

Whhy use so many digits in yy axis? Explain.

4.2. Results Analysis”: is it results and discussion then (Analysis…)? Revise…

This is not the place to add “new” formulas:

“The fertilizer application and per capita gross agricultural production(???)decoupling index”

And previously “gross agricultural production value(????”

[Once defined, earlier on…, only nomenclature must be used…]

There is some “collision” in terms…

See the remaining…

5. Conclusions and prospect” again “prospect” needs revision…

I would like to see the following structure:

Brief contextualization

Brief methodology

Main findings

Practical implications

(similar, but more clear than abstract…)

Better connect abstract and conclusions sections

Although understanding the aim of the text, I do believe the methodology is not clear nor assertively addressed and authors do need to work on the entire text and better explore tables and figures (very poor) in terms of making the text relevant.

References: better analyse instructions:

https://www.mdpi.com/journal/ijerph/instructions

“References should be described as follows, depending on the type of work:…”

Update references, more from 2019…

Author Response

Manuscript ID: ijerph-593479

Title: Analysis of the Agricultural Economy and Agricultural Pollution in Chengdu, Sichuan Province, China

Dear Editors and Reviewers: 

Thank you for your letter and for the reviewers’ comments concerning our manuscript. Those comments are all valuable and very helpful for revising and improving our paper, as well as the important guiding significance to our research. We have studied comments carefully and have made correction which we hope meet with approval.

The main corrections in the paper and the responds to the reviewer’s comments are marked in red and italic as following:

Title: I believe authors must revise the term “Decoupling”

[See that “decoupling index theory” appears ONCE in the text, how can it be chosen as keyword? Revise keywords…]

Abstract: The abstract needs to be clear in terms of results and findings, namely explicitly clarifying meaning and ranges, namely in terms of index and quantitative data.

Reading the abstract it is not clear what kind if index and how it works…

Add practical implications

Revise non clear or scientific expressions: “Overall,”, “relatively healthy” 

Revise keywords…, so they better reflect the manuscript. 

Modification explanation: Thank you for detailed advice. We have revised the title and removed "Decoupling". As for your opinion on keywords, we think it is very useful. We have replaced "decoupling index theory" with "decoupling index". We revised the abstract and replaced the previous description of the results with intuitive research data to make the research results more clearly reflected in the abstract. We have revised the wording of unclear words to make the conclusion easier to understand.

Analysis of the Agricultural Economy and Agricultural Pollution in Chengdu, Sichuan Province, China (line 2-4 under revised manuscript).

It was found that the decoupling index for pesticide use and per capita gross agricultural production showed "strong decoupling" ideal states in six years and the chemical fertilizer use was found to have a "strong decoupling" ideal state in five years; however, in the past four years, agricultural plastic film has shown a state of "weak decoupling", which directly leads to the whole improvement of agricultural pollution. The research results show that Chengdu's agricultural economy and agricultural pollution were found to have a healthy and coordinated development. (line 18-24 under revised manuscript).

Keywords:  agricultural economy; agricultural pollution; decoupling index (line 25 under revised manuscript).

Revise all spacing… “people[1].”

Modification explanation: Your examination is very careful. We have checked and revised all the similar problems in this paper.

Where are the references in every simiar case? “this paper used the Tapio decoupling index”

Modification explanation: Reference as follows have been added to this part.

Tapio P. Towards a theory of decoupling: degrees of decoupling in the EU and the case of road traffic in Finland between 1970 and 2001J].Transport Policy200512( 2) : 137 151(line 47 under revised manuscript).

Why “decoupling index” and then  also “"decoupling" index”?

Modification explanation: Your examination is very careful. We apologize for our carelessness in writing and have corrected the problem.

The decoupling index has usually been used to measure the relationships and non-synchronous changes between resource consumption, environmental pressure, and economic growth [15].  (line 114-115 under revised manuscript)

Consider revising the way references are presented in the text, there are times when direct referencing style would also be considered not just at the end… “[]”

Modification explanation: Your suggestion is very good. We have made some adjustments in this paper. In sentences involving scholars' opinions, we usually add "[]" symbols after the name of scholars.   

Qiu [7] believes that promoting the coordinated development of economy and environment is the core of urban sustainable development, so he decoupled the economic growth and resource environment of coal cities. Zhang [8, 9] and Zhou [10] believed that China has paid a lot of environmental costs in the past development process. Energy conservation and emission reduction are the top priority. They all used decoupling indices to study economic growth and resource consumption and economic growth and energy consumption. Yi [11] studied tourism economic growth and ecological environmental pressure issues in geoparks. After studying the economic growth and construction land Liu [12] believed that the dependence of economic growth on construction land resources is gradually decreasing, and manufacturing industry is an important industry to stimulate China's economy.  (line 59-68 under revised manuscript)

Through Zhang's analysis [14], it is concluded that the sustainability of China's economic growth increases with economic growth. Long [15] refers to resource consumption and environmental pollution as ecological loss and analyzed the economic growth, ecological loss and welfare levels in the process of urbanization.  (line 71-74 under revised manuscript)

Xiang [23] studied the urbanization and the associated carbon intensity and concluded that urbanization was rising and carbon emissions were declining. (line 85-86 under revised manuscript)

References? “Chengdu Statistical Yearbooks.” See all similar cases It mentioned and existing, then cite…

Modification explanation: We apologize for our previous negligence in citing references, and we have added references to this part.

To establish the agricultural economic growth and agricultural pollution research model and select the specific indicators, this paper synthesized previous studies on the relationships between economic growth and the ecological environment or ecological pollution, and selected indicators from 2008 to 2017 from the Chengdu Statistical Yearbooks [32]. (line 102-105 under revised manuscript)

Therefore, to explore the specific relationships between Chengdu's economic development and its environmental protection, based on data from the Chengdu Statistical Yearbooks [32] from 2008 to 2017, (line 132-134 under revised manuscript)

Chengdu Statistical Bureau. Chengdu Statistical Yearbook; China Statistics Press: Beijing, China, 2008–2017. (line 368 under revised manuscript)

7.To me, this is not the place to add considerations regarding novelty…

“Therefore, the innovations in this study were as 79 follows.”

And please avoid the “listing”…

Modification explanation: We changed the position of innovation to the introduction. In the introduction, the innovation points are pointed out directly, and the research objectives and significance of this paper are highlighted. We no longer use the “listing” style in writing about innovation.

Thus, the main innovations of this paper are as follows. The relationship between economic growth and ecological pollution was studied from an agricultural perspective using a decoupling index to explore the key factors affecting the decoupling effect and to identify the direct relationships between the agricultural economy and agricultural pollution. Agricultural environmental pollution indicators were selected; pesticides, agricultural fertilizers and agricultural plastic film; which better reflected the characteristics of agricultural pollution. (line 49-55 under revised manuscript).

8.References?

“The Organization 95 for Economic Cooperation and Development (OECD) has generally used the decoupling index in its 96 agricultural policy research, and has divided the "decoupling" into absolute decoupling and relative 97 decoupling.”

See similar situations

References? “Taking the OECD decoupling index as a reference,”

Modification explanation: Once again, we apologize for the negligence in the citation of references, and we have added references to all relevant parts of the article.

The Organization for Economic Cooperation and Development (OECD) has generally used the decoupling index in its agricultural policy research [33], (line 117-119 under revised manuscript).

Taking the OECD decoupling index [33] as a reference, a decoupling index measurement model was constructed to assess the agricultural economy and the associated agricultural pollution, the specific calculation steps and technical roadmap for which are shown in Figure 1.  (line 155-157 under revised manuscript).

OECD. Effects of quantitative constraints on the degree of decoupling of crop support measures [R]. Paris: OECD, 2005. (line 369-370 under revised manuscript).

Revise typos… “healthy..”

Revise italics in references: “average [4].”

Modification explanation: Thank you for reading and checking our articles so carefully. We have adjusted the format.

Per capita gross agricultural production was selected to reflect the economic growth in agriculture as it can reveal the total scale of regional agricultural production and economic growth. The indicators selected to reveal the agricultural pollution problems were the use of pesticides, agricultural fertilizers and agricultural plastic film. China is the world's largest pesticide user, with the pesticide use area being 2.5 times the world average [5]. China uses 40% of the world's chemical fertilizers; however, as the organophosphorus, organochlorines, heavy metals and other toxins from pesticides, herbicides and some fertilizers can pollute water bodies [4], pesticide pollution is a significant agricultural pollution source and its index has important reference value. The excessive use of chemical fertilizers can also seriously pollute surface and groundwater. Therefore, chemical fertilizers are another major source of agricultural pollution. The extensive use of agricultural film in agricultural production has been found to accelerate the "death" of cultivated land, and its degradation releases carcinogenic dioxins into the air, endangering life and health [5]. In the past literature[34, 35], the above indicators are also used to measure agricultural environmental pollution. Therefore, this paper used agricultural plastic film as another index to evaluate agricultural pollution. (line 140-153 under revised manuscript).

Figures: PLEASE use proper caption (self-explanatory ones…): “Figure 1. Technical roadmap” says nothing…

Figure is distorted, check consistency and do not use font bigger than text…

Language inside needs revision

Modification explanation: We redrawn the technology roadmap and named Figure 1 "Technical roadmap for decoupling analysis of agricultural economy and agricultural pollution." This makes it easier for readers to understand what Figure 1 is intended to express. We have adjusted some of the text expression inside the figure to make the content of the whole technology roadmap more accurate.

Figure 1. Technical roadmap for decoupling analysis of agricultural economy and agricultural pollution. (line 158-160 under revised manuscript).

Why referring “original data” in “3.3.1. Calculating the Variation Values for the Indicators in the Original Data”. [used four times in the text]

Modification explanation: The first step is to use the original data to calculate the change rate of each index when we calculate the decoupling index. According to your suggestion, we have revised the title to make it look more concise.

2.3.1. The Calculating Process and Formula of Decoupling Index (line 161 under revised manuscript).

To me, the language needs clarification and assertiveness: “First, based on the selected indexes,”

Modification explanation: We rewrite this as” Based on the selected indicators……”.

Based on the selected indicators, the change values in each index in the original data in each year were calculated and compared with the previous year. (line 177-178 under revised manuscript).

Why use “CNY” in an international indexed Journal? 

Check international unit system: “Use/Ton” and then “Application / Tonnes”. Units must be indicated inside ()

Modification explanation: "CNY" is usually the unit of RMB in the paper. The unit of RMB used in the following article is "CNY". In addition, thank you for checking the unit problem in our table. We have revised the spelling problem and bracketed the unit.

Xie, X. and L. Pu, Assessment of Urban Ecosystem Health Based on Matter Element Analysis: A Case Study of 13 Cities in Jiangsu Province, China. International Journal of Environmental Research and Public Health, 2017. 14(8): p. 940.

Change or changed? “Table 2. Change value” 

Caption should consider including time period 

Then cannot be a second a new subsection, not me:

“3.3.2. Calculating the Change Rate of Each Index in Each Year 143

Second, the annual”

Modification explanation: Thank you for checking the tense of the language in the title. We have revised this issue. We have reorganized the structure of the whole calculation part, removed the title you said, and made the calculation process look smoother.

Table 2. Changed value for the original data for the selected indicators in each year compared with the previous year. (line 181-182 under revised manuscript).

To me mathematical data must be presented (and removed from table using both definition and formula at the same time…) otherwise, as equations (and numbered, au usually…), and terms (parameters) defined and units presented, where available 

There are ways to integrate in a vertical table… “Table 4. Calculated decoupling index.”, just further work it…

Modification explanation: At the beginning of the calculation part, we add a formula description of the whole calculation process, explaining in detail the meaning of each symbol or term. In this way, the reader can have a detailed understanding of the calculation process and symbolic terms at the beginning, and will not produce ambiguity in the use of symbols or terms later.

The formula for calculating the decoupling index is as follows:

Where:  indicates the decoupling Index for the agricultural economy and agricultural pollution.  and indicate the per capita gross agricultural production value and agricultural pollution indicators respectively.  and  indicate the change value for the per capita gross agricultural production value and agricultural pollution indicators respectively.  and  indicate the change rate of per capita gross agricultural production value and the change rate of agricultural pollution indicators respectively.

The value of  is 1, 2 or 3, when  = 1 indicates the decoupling index for pesticide use and per capita gross agricultural production indicates the pesticide usewhen  = 2 indicates the decoupling index for the amount of fertilizer used in agriculture and per capita gross agricultural production indicates the agricultural fertilizer applicationwhen  = 3 indicates the decoupling index for agricultural plastic film use and per capita gross agricultural production value indicates the agricultural plastic film usage. (line 161-175 under revised manuscript).

The use of so many headings to breaks the text…

And I do feel the language must be clarified:

“4.1. Annual Decoupling Index State Classification for Each Index”

In tables, use just the nomenclature and notes can give additional information, including the identification... “Decoupling Index ?”.

Modification explanation: As indicated in Questions 14 and 15, we have revised the structure and subtitle of this section. At the beginning of the calculation part, we write out the calculation process and definition of symbolic terms, so that we can use the letter T to express the decoupling index.

3.1. Decoupling Index State Classification (line 195 under revised manuscript).

The formula for calculating the decoupling index is as follows:

Where:  indicates the decoupling Index for the agricultural economy and agricultural pollution.  (line 162-165 under revised manuscript).

The value of  is 1, 2 or 3, when  = 1 indicates the decoupling index for pesticide use and per capita gross agricultural production indicates the pesticide usewhen  = 2 indicates the decoupling index for the amount of fertilizer used in agriculture and per capita gross agricultural production indicates the agricultural fertilizer applicationwhen  = 3 indicates the decoupling index for agricultural plastic film use and per capita gross agricultural production value indicates the agricultural plastic film usage.  (line 170-175 under revised manuscript).

No formatted (justified…) text here: “Based on the TAPIO decoupling”

[and why previously lower letter, still no references… “Tapio decoupling”].

Modification explanation: We set up the format alignment of this paragraph and unified the writing of "Tapio decoupling". Thank you again for careful examination.

Based on the Tapio decoupling index classification[6] in Table 5, the decoupling index calculations in Table 4 were corresponded one by one, and the specific classifications for each decoupling index in each year determined, as shown in Table 6. (line 201-203 under revised manuscript).

Figure 2: No xx axis label…

Whhy use so many digits in yy axis? Explain.

Modification explanation: We redrawn Figure 2 to make the x-axis clearer. We use six decimals on the y-axis in order to correspond calculation process which use six decimals in the, so that the results of the figure can be presented more accurately.

Figure 2. Annual decoupling index diagram (line 212-213 under revised manuscript).

4.2. Results Analysis”: is it results and discussion then (Analysis…)? Revise…

This is not the place to add “new” formulas:

“The fertilizer application and per capita gross agricultural production(???)decoupling index”

And previously “gross agricultural production value(????”

[Once defined, earlier on…, only nomenclature must be used…]

There is some “collision” in terms…

See the remaining…

Modification explanation: Your suggestion makes this article more precise in description. We revised the subtitle and used full names instead of symbols for the discussion and analysis of the results. In the table, we use uniformly defined symbols or terms to describe the indicators.

3.2. Result Discussion and Analysis (line 214 under revised manuscript).

5. Conclusions and prospect” again “prospect” needs revision… 

I would like to see the following structure:

Brief contextualization

Brief methodology

Main findings

Practical implications

(similar, but more clear than abstract…)

Better connect abstract and conclusions sections

Modification explanation: We revised the title. We have adjusted the conclusions, highlighted the method, objectives and research significance of this paper once again, including the innovation and advantages of this paper. We generalize more briefly in language, and the content of the conclusion corresponds to the revised abstract.

Conclusions

This paper used the decoupling index to study the relationship between economic growth and environmental pollution from agricultural perspective in Chengdu, a core agricultural city in Western China. A new model for measuring agricultural economy and agricultural pollution is proposed and validated with the data from 2008 to 2017. The following conclusions were drawn from the analyses. (line 249-254 under revised manuscript).

Other Modifications:

We combined the introduction with the literature review. Tables and figures are now are a lot clearer to read and easier to understand.

We tried our best to improve the manuscript and made some changes in the manuscript. These changes will not influence the content and framework of the paper.

We appreciate for Editors/Reviewers’ warm work earnestly, and hope that the correction will meet with approval.

Once again, thank you very much for your comments and suggestions.

Sipan Li

Sichuan University

2019-9-20

Round 2

Reviewer 1 Report

Title: Now it's too generic! It should be better "Analysis of the Agricultural Economy and Agricultural Pollution through (or using) the decoupling index in Chengdu, Sichuan Province, China".

Summary: you should not report the results as you got them, but you should write the results from the analysis in a qualitative way. After explaining the decoupling index, you should find a more discursive way of reassessing the results.

Introduction: This version is better than the previous one, but there are still too many repetitions! Also, all the studies could be presented in a better way.

Lines 35-40 you should reformulate the sentence. The period is too long.

Lines 40-41 and 41-42: you should reformulate the sentence by combining the 2 periods.

Lines 47-48: "to examine the relationships between the agricultural economy and agricultural pollution ..." is a repetition. It would be better "to examine this relationship and provide a new model ..."

Lines 53-55: I don't like the way the sentence is formulated.

Lines 56-57: "domestic (???)" you should use a synonym.

Line 66: "construction land" It would be better "building plot" or "building land".

In general, a language revision is absolutely necessary to make text reading clearer and more fluid.

Materials and Methods: please, remove some repetitions.

Table 5. It is not necessary to repeat the word "decoupling" in each row of the "status" column because there is already the column with "Negative Decoupling", "Decoupling" and "Connection".

The rest is better than before.

Author Response

Manuscript ID: ijerph-593479

Title: Analysis of the Agricultural Economy and Agricultural Pollution in Chengdu, Sichuan Province, China

Dear Editors and Reviewers: 

Thank you for your letter and for the reviewers’ comments concerning our manuscript. Those comments are all valuable and very helpful for revising and improving our paper, as well as the important guiding significance to our research. We have studied comments carefully and have made correction which we hope meet with approval.

The main corrections in the paper and the responds to the reviewer’s comments are marked in red and italic as following:

Title: Now it's too generic! It should be better "Analysis of the Agricultural Economy and Agricultural Pollution through (or using) the decoupling index in Chengdu, Sichuan Province, China".

Modification explanation: Thank you for careful examination of the article. We have revised the title according to your opinion.

Analysis of the Agricultural Economy and Agricultural Pollution through the decoupling index in Chengdu, Sichuan Province, China. (line 2-4 under revised manuscript).

Summary: you should not report the results as you got them, but you should write the results from the analysis in a qualitative way. After explaining the decoupling index, you should find a more discursive way of reassessing the results.

Modification explanation: We adopted a qualitative approach to analyze the results. In addition, we describe the practical significance of the findings more clearly in the abstract.

It was found that the decoupling index for pesticide use and per capita gross agricultural production had a large number of "strong decoupling" ideal states and half the chemical fertilizer use was found to have a "strong decoupling" ideal state; however, the increased use of plastic film has resulted in increased agricultural pollution. The research results show that growing with the development of Chengdu's agricultural economy, the use of pesticides and fertilizers has been decreasing. Chengdu's agricultural economy and agricultural pollution were found to have a healthy and coordinated development. The development of agricultural economy has not caused much damage to the agricultural environment. In addition, the reduction of agricultural pollution has also played a good role in public health. (line 18-27 under revised manuscript).

Introduction: This version is better than the previous one, but there are still too many repetitions! Also, all the studies could be presented in a better way.

Modification explanation: Based on your valuable comments, we have revised many sentences in the introduction. Rewritten many repetitive sentences and highlighted the significance of this study and the issues of concern.

Agriculture is the foundation of the Chinese national economy as it provides materials for both residents and other industry sectors. As China is a large agricultural country, agriculture is vital to both the Chinese government and the people [1]. The agricultural economy is made up of agricultural economic relationships and agricultural economic activities. Since the Economic Reform and open up, China's agricultural economy has achieved great success with most problems that had been associated with insufficient agricultural product supply being solved. In 2017, China's total grain output reached 617.91 million tonnes [2], twice the total grain output of 304.765 million tonnes in 1978. However, the measures taken to increase agricultural production have resulted in the excessive use of chemical fertilizers and pesticides can no longer meet the requirements of the development of agricultural economy in China under the new situation [3]. Because of the increase in agricultural pollution; chemical fertilizer pollution, pesticide pollution and intensive farm pollution; which has resulted in nitrogen, phosphorus, pesticides and other organic or inorganic pollutant entering the surface water, groundwater, and soil environments through surface runoff [4]. All these pollutions endanger the environment and the health of residents [5]. Modern agriculture depends heavily on pesticide use and chemical fertilizer, which have successfully increased roductivity but led to increasing concerns regarding the environment and human health [6]. Therefore, it is necessary to study the relationship between economic growth and environmental pollution as the destruction of the ecological environment can restrict economic development. Chengdu is a core city in the southwest of China. It has a good level of agricultural development and a high agricultural status in the western region. However, there is no analysis of the relationship between the agricultural economy and the agricultural environment in Chengdu in the previous literature. In order to better measure the relationship between agricultural economy and agricultural pollution, realize the sustainable development of the agricultural economy and explore public health issues, this paper use decoupling index [7] to examine this relationships and provide a new model for the study of economic growth and environmental pollution. (line 31-55 under revised manuscript).

Lines 35-40 you should reformulate the sentence. The period is too long.

Modification explanation: We have separated this sentence to make it smoother and easier for the reader to understand its meaning.

However, the measures taken to increase agricultural production have resulted in the excessive use of chemical fertilizers and pesticides can no longer meet the requirements of the development of agricultural economy in China under the new situation [3]. Because of the increase in agricultural pollution; chemical fertilizer pollution, pesticide pollution and intensive farm pollution; which has resulted in nitrogen, phosphorus, pesticides and other organic or inorganic pollutant entering the surface water, groundwater, and soil environments through surface runoff [4]. All these pollutions endanger the environment and the health of residents [5]. (line 38-44 under revised manuscript)

Lines 40-41 and 41-42: you should reformulate the sentence by combining the 2 periods.

Modification explanation: Based on your suggestion, we revised the sentence.

Modern agriculture depends heavily on pesticide use and chemical fertilizer, which have successfully increased roductivity but led to increasing concerns regarding the environment and human health [6]. Therefore, it is necessary to study the relationship between economic growth and environmental pollution as the destruction of the ecological environment can restrict economic development. (line 44-48 under revised manuscript)

Lines 47-48: "to examine the relationships between the agricultural economy and agricultural pollution ..." is a repetition. It would be better "to examine this relationship and provide a new model ..."

Modification explanation: Thank you for suggestions. We have rewritten this sentence in accordance with your opinion.

In order to better measure the relationship between agricultural economy and agricultural pollution, realize the sustainable development of the agricultural economy and explore public health issues, this paper use decoupling index [7] to examine this relationships and provide a new model for the study of economic growth and environmental pollution. (line 51-55 under revised manuscript)

Lines 53-55: I don't like the way the sentence is formulated.

Modification explanation: We have adjusted the way of writing the innovative points to make the sentences more clear.

The relationship between economic growth and ecological pollution was studied from an agricultural perspective in Chengdu, a core agricultural city in Western China, using a decoupling index to explore the key factors affecting the decoupling effect and to identify the direct relationships between the agricultural economy and agricultural pollution. A new model for measuring agricultural economy and agricultural pollution is proposed and validated with the data from 2008 to 2017. (line 276-281 under revised manuscript)

Lines 56-57: "domestic (???)" you should use a synonym.

Modification explanation: We no longer use this word. We used a more concise way to express this sentence.

With the further deepening of the global resource and environmental crisis, economic, energy and environmental issues have become the focus of scholars. (line 56-57 under revised manuscript)

Line 66: "construction land" It would be better "building plot" or "building land".

Modification explanation: Your expression is obviously more accurate. We have revised it.

After studying the economic growth and building land (line 66 under revised manuscript)

In general, a language revision is absolutely necessary to make text reading clearer and more fluid.

Modification explanation: We listen to your suggestions and adjust the writing style in the text and make the text reading clearer and more fluid.

Materials and Methods: please, remove some repetitions.

Modification explanation: We carefully examined the possible repetitive sentences and rewrote the following sentences.

Therefore, to explore the specific relationships between Chengdu's economic development and its environmental protection, based on data from the Chengdu Statistical Yearbooks [33] from 2008 to 2017, this paper selected the per capita gross agricultural production value, and pesticide, agricultural fertilizer, and agricultural plastic use to calculate the decoupling index. (line 131-134 under revised manuscript)

Table 5. It is not necessary to repeat the word "decoupling" in each row of the "status" column because there is already the column with "Negative Decoupling", "Decoupling" and "Connection".

Modification explanation: Thank you for advice. We have revised the contents of the table.

Table 5. Classification of decoupling index. (line 200 under revised manuscript)

Other Improvements:

Public health is discussed in this paper. We separate the discussion and analysis of the results so that the structure of the article is better.

We tried our best to improve the manuscript and made some changes in the manuscript. These changes will not influence the content and framework of the paper.

We appreciate for Editors/Reviewers’ warm work earnestly, and hope that the correction will meet with approval.

Once again, thank you very much for your comments and suggestions.

Sipan Li

Sichuan University

2019-10-4

Reviewer 3 Report

Highlight changes in yellow in a next revision, please. No track changes.

Consider comments in the entire text.

Please see that the reviewer’s time is valuable and I specifically asked revisions to be indicated in yellow, rather than detailing in answers, so I would be able to focus on changes…

Please answer to comments one by one, and not in bulk…

Abstract: I always have difficulty in analysing an abstract where three is no quantitative data [also in Conclusions section…] resulting from the data presented in the text… I would like to see more assertiveness in that, namely concluding with “research results show that Chengdu's agricultural economy and 23 agricultural pollution were found to have a healthy and coordinated development. 24” and linking to practical implications.

Revise spacing all over “The research results show that Chengdu's agricultural economy”, at least in the PDF

Check the entire text then for reference lack:

“Depending on the differences in the elastic coefficient, the Tapio tool”

“3. Where are the references in every similar case? “this paper used the Tapio decoupling index”

Modification explanation: Reference as follows have been added to this part.

Tapio P. Towards a theory of decoupling: degrees of decoupling in the EU and the case of road traffic in Finland between 1970 and 2001J].Transport Policy200512( 2) : 137 151(line 47 under revised manuscript).”

I see “Chengdu Statistical Yearbooks” Plural, and then, one reference, and then reference 32 reference, singular… No coherence…

Just to illustrate it compromises the text…

“6. References? “Chengdu Statistical Yearbooks.” See all similar cases It mentioned and existing, then cite…

Modification explanation: We apologize for our previous negligence in citing references, and we have added references to this part.

(…)”

Again, the innovation may be at discussion and conclusions, not in the middle of the Introduction…

“7.To me, this is not the place to add considerations regarding novelty…

“Therefore, the innovations in this study were as 79 follows.”

And please avoid the “listing”…

Modification explanation: We changed the position of innovation to the introduction. In the introduction, the innovation points are pointed out directly, and the research objectives and significance of this paper are highlighted. We no longer use the “listing” style in writing about innovation.

Thus, the main innovations of this paper are as follows. The relationship between economic growth and ecological pollution was studied from an agricultural perspective using a decoupling index to explore the key factors affecting the decoupling effect and to identify the direct relationships between the agricultural economy and agricultural pollution. Agricultural environmental pollution indicators were selected; pesticides, agricultural fertilizers and agricultural plastic film; which better reflected the characteristics of agricultural pollution. (line 49-55 under revised manuscript).”

Further work on captions…

What reality? What context?

Figure 1. Technical roadmap for decoupling analysis of agricultural economy and agricultural 159 pollution.”

“10. Figures: PLEASE use proper caption (self-explanatory ones…): “Figure 1. Technical roadmap” says nothing…

Figure is distorted, check consistency and do not use font bigger than text…

Language inside needs revision

Modification explanation: We redrawn the technology roadmap and named Figure 1 "Technical roadmap for decoupling analysis of agricultural economy and agricultural pollution." This makes it easier for readers to understand what Figure 1 is intended to express. We have adjusted some of the text expression inside the figure to make the content of the whole technology roadmap more accurate.

Figure 1. Technical roadmap for decoupling analysis of agricultural economy and agricultural pollution. (line 158-160 under revised manuscript).”

Is it raw data then?

“11. Why referring “original data” in “3.3.1. Calculating the Variation Values for the Indicators in the Original Data”. [used four times in the text]

Modification explanation: The first step is to use the original data to calculate the change rate of each index when we calculate the decoupling index. According to your suggestion, we have revised the title to make it look more concise.

2.3.1. The Calculating Process and Formula of Decoupling Index (line 161 under revised manuscript).”

(…)

Decoupling term in every cell unnecessary…, never seen such a thing…

Table 6. Status of the decoupling index for each indicator.”

The same here:

“Depending on the differences in the elastic coefficient, the Tapio tool divides the decoupling 196 state into eight types: strong decoupling, strong negative decoupling, weak decoupling, weak 197 negative decoupling, expanding negative decoupling, expanding connection, recession decoupling 198 and recession connection[11].”

And check spacing again, before references…

Is it a formula or an equation, then? And it must be numbered…

“The formula for calculating the decoupling index is as follows:”

I believe authors must consider further wring the text so it may be relevant…

Figure 2: all axis must have a legend (T1, T2, T3 means what, so the reader may easily check? Also in Table 4, as example, consider adding notes below tables)

Then highlight accuracy…

“18. Figure 2: No xx axis label…

Why use so many digits in yy axis? Explain.

Modification explanation: We redrawn Figure 2 to make the x-axis clearer. We use six decimals on the y-axis in order to correspond calculation process which use six decimals in the, so that the results of the figure can be presented more accurately.

(…)

Again, do discussion come before analysis, it should be simultaneous, call it results and discussion…

3.2. Result Discussion and Analysis

“19. “4.2. Results Analysis”: is it results and discussion then (Analysis…)? Revise…

(…)”

This is not a contextualization, thus justifying the need for this study…

4. Conclusions 250

This paper used the decoupling index to study the relationship between economic growth and 251 environmental pollution from agricultural perspective in Chengdu, a core agricultural city in 252 Western China.”

“20. “5. Conclusions and prospect” again “prospect” needs revision… 

I would like to see the following structure:

Brief contextualization

Brief methodology

Main findings

Practical implications

(similar, but more clear than abstract…)

Better connect abstract and conclusions sections”

See that expressions as “was relatively poor.” Must be revised. Where is clarity.

See that all this must change…

“279 Author Contributions: For research articles with several authors, a short paragraph specifying their individual contributions must be provided. The following statements should be used “conceptualization, X.X. and Y.Y.; 280 methodology, X.X.; software, X.X.; validation, X.X., Y.Y. and Z.Z.; formal analysis, X.X.; investigation, X.X.; 281 resources, X.X.; data curation, X.X.; writing—original draft preparation, X.X.; writing—review and editing, X.X.; 282 visualization, X.X.; supervision, X.X.; project administration, X.X.; funding acquisition, Y.Y.”, please turn to the 283 CRediT taxonomy for the term explanation. Authorship must be limited to those who have contributed 284 substantially to the work reported. 285

Funding: Please add: “This research received no external funding” or “This research was funded by NAME OF 286 FUNDER, grant number XXX” and “The APC was funded by XXX”. Check carefully that the details given are 287 accurate and use the standard spelling of funding agency names at https://search.crossref.org/funding, any 288 errors may affect your future funding. 289

Acknowledgments: In this section you can acknowledge any support given which is not covered by the author 290 contribution or funding sections. This may include administrative and technical support, or donations in kind 291 (e.g., materials used for experiments). 292

Conflicts of Interest: Declare conflicts of interest or state “The authors declare no conflict of interest.” Authors 293 must identify and declare any personal circumstances or interest that may be perceived as inappropriately 294 influencing the representation or interpretation of reported research results. Any role of the funders in the design 295 of the study; in the collection, analyses or interpretation of data; in the writing of the manuscript, or in the 296 decision to publish the results must be declared in this section. If there is no role, please state “The funders had 297 no role in the design of the study; in the collection, analyses, or interpretation of data; in the writing of the 298 manuscript, or in the decision to publish the results”.”

Many of the aspects addressed are not the goal of reviewing… Authors must previously revise the manuscript carefully.

I ask authors to consider looking at the text from an independent perspective, analysing all aspects in a global manner, particularly in connecting the subject to the scope of this journal, called… “Int. J. Environ. Res. Public Health”. Heath is not really discussed, rather sustainability…

Check the end…

“The relationship between the regional agricultural economic development and sustainable 275 regional agricultural development needs to be explored to determine the impact of the changes in the 276 regional agricultural economy and agricultural pollution on regional education. The influencing 277 factors for the changes in the agricultural pollution indicators also need to be further explored.”

Author Response

Manuscript ID: ijerph-593479

Title: Analysis of the Agricultural Economy and Agricultural Pollution in Chengdu, Sichuan Province, China

Dear Editors and Reviewers: 

Thank you for your letter and for the reviewers’ comments concerning our manuscript. Those comments are all valuable and very helpful for revising and improving our paper, as well as the important guiding significance to our research. We have studied comments carefully and have made correction which we hope meet with approval.

The main corrections in the paper and the responds to the reviewer’s comments are marked in red and italic as following:

Please see that the reviewer’s time is valuable and I specifically asked revisions to be indicated in yellow, rather than detailing in answers, so I would be able to focus on changes…

Modification explanation: We apologize for forgetting to mark the text in yellow last time. We are very sorry that our negligence has caused you trouble in reviewing manuscripts. This time, we marked the revised parts in yellow. We hope that this will be helpful for your review.

Abstract: I always have difficulty in analysing an abstract where three is no quantitative data [also in Conclusions section…] resulting from the data presented in the text… I would like to see more assertiveness in that, namely concluding with “research results show that Chengdu's agricultural economy and 23 agricultural pollution were found to have a healthy and coordinated development. 24” and linking to practical implications.

Modification explanation: We have adjusted the abstract according to your opinion. We described more clearly the practical significance of the results.

It was found that the decoupling index for pesticide use and per capita gross agricultural production had a large number of "strong decoupling" ideal states and half the chemical fertilizer use was found to have a "strong decoupling" ideal state; however, the increased use of plastic film has resulted in increased agricultural pollution. The research results show that growing with the development of Chengdu's agricultural economy, the use of pesticides and fertilizers has been decreasing. Chengdu's agricultural economy and agricultural pollution were found to have a healthy and coordinated development. The development of agricultural economy has not caused much damage to the agricultural environment. In addition, the reduction of agricultural pollution has also played a good role in public health. (line 18-27 under revised manuscript).

Revise spacing all over “The research results show that Chengdu's agricultural economy”, at least in the PDF.

Modification explanation: We carefully revised sentences.

The research results show that growing with the development of Chengdu's agricultural economy, the use of pesticides and fertilizers has been decreasing. (line 22-23 under revised manuscript).

Check the entire text then for reference lack: “Depending on the differences in the elastic coefficient, the Tapio tool”

Modification explanation: Thank you for your comments. Your examination is very careful. We have supplemented the references.

Depending on the differences in the elastic coefficient, the Tapio tool [7] divides the decoupling state into eight types: (line 196-197 under revised manuscript)

I see “Chengdu Statistical Yearbooks” Plural, and then, one reference, and then reference 32 reference, singular… No coherence…

Just to illustrate it compromises the text…

Modification explanation: Thank you for your comments. In the references to the "Chengdu Statistical Yearbooks", we have marked the years as "2008-2017", indicating that we have used the Yearbook information for ten consecutive years Yearbooks.

Again, the innovation may be at discussion and conclusions, not in the middle of the Introduction…

Modification explanation: According to your suggestion, we have written the innovation point in the conclusion and adjusted the content of it.

This paper is devoted to exploring the relationship between agricultural economic development and agricultural pollution, and discussing whether it endangers public health. Therefore, the main innovations of this paper are as follows. The relationship between economic growth and ecological pollution was studied from an agricultural perspective in Chengdu, a core agricultural city in Western China, using a decoupling index to explore the key factors affecting the decoupling effect and to identify the direct relationships between the agricultural economy and agricultural pollution. A new model for measuring agricultural economy and agricultural pollution is proposed and validated with the data from 2008 to 2017. (line 274-281 under revised manuscript)

Further work on captions…What reality? What context?

Modification explanation: Figure 1 is to show the steps of this study and make it easier for readers to understand the process of this study. Referring to other articles on IJERPH, we now name Figure 1 as "Research framework for decupling analysis of agricultural economy and agricultural pollution".

Figure 1. Research framework for decoupling analysis of agricultural economy and agricultural pollution. (line 155-156 under revised manuscript).

Is it raw data then?

Modification explanation: Yes, it does mean "raw data". We have replaced all "original data" with "raw data".

Based on the selected indicators, the change values in each index in the raw data in each year were calculated and compared with the previous year. The change values for the raw data and each index and the comparisons with the previous year are shown in Tables 1 and 2. (line 173-175 under revised manuscript).

Decoupling term in every cell unnecessary…, never seen such a thing…

“Table 6. Status of the decoupling index for each indicator.”

The same here: “Depending on the differences in the elastic coefficient, the Tapio tool divides the decoupling 196 state into eight types: strong decoupling, strong negative decoupling, weak decoupling, weak 197 negative decoupling, expanding negative decoupling, expanding connection, recession decoupling 198 and recession connection[11].”

And check spacing again, before references…

Modification explanation: We removed the unnecessary "decoupling" in Table 6 and that paragraph. The blank space was added before the references.

Depending on the differences in the elastic coefficient, the Tapio tool [7] divides the decoupling state into eight types: strong, strong negative, weak decoupling, weak negative, expanding negative, expanding connection, recession decoupling and recession connection [12]. (line 196-199 under revised manuscript).

Table 6. Status of the decoupling index for each indicator. (line 204 under revised manuscript).

Is it a formula or an equation, then? And it must be numbered…

Modification explanation: This is a formula. Your reminder is very good. We have numbered the formula.

 (line 159 under revised manuscript).

Figure 2: all axis must have a legend (T1, T2, T3 means what, so the reader may easily check? Also in Table 4, as example, consider adding notes below tables)

Modification explanation: In Chapter 2.3.1, the meanings of T1, T2 and T3 are explained in detail. According to your request, we have commented on each table and figure that involves these three terms.

Note: T1 indicates the decoupling index for pesticide use and per capita gross agricultural production. T2 indicates the decoupling index for the amount of fertilizer used in agriculture and per capita gross agricultural production. T3 indicates the decoupling index for agricultural plastic film use and per capita gross agricultural production value. (line 190-193, 205-208 under revised manuscript).

Then highlight accuracy…

Modification explanation: Marking the data directly on the figure may make the figure look confusing, thus we draw the Figure 2, Figure 3 and Figure 4 to show the results of the study respectively, and mark the specific data of each year in the figure.

Figure 2. Annual decoupling index for pesticide use and per capita gross agricultural production (line 224-226 under revised manuscript).

Figure 3. Annual decoupling index for the amount of fertilizer used in agriculture and per capita gross agricultural production (line 238-240 under revised manuscript).

Figure 4. Annual decoupling index for agricultural plastic film use and per capita gross agricultural production (line 251-253 under revised manuscript).

Again, do discussion come before analysis, it should be simultaneous, call it results and discussion… 

Modification explanation: We have adopted your suggestion. We divide this part into results discussion and results analysis. In the discussion of the results, we insert Figures 2, 3 and 4 into the text and discuss the results separately.

3.2. Results and Discussion

Pesticide use contributes to increased agricultural production and improved product quality, but overuse or abuse of pesticides has been related to negative externalities such as human health effects and external ecological effects [6]. Heavy metals may be added to soils in agricultural fertilizers and pesticides, soil amendments, or organic fertilizers [37].

The decoupling index for Chengdu from 2009 to 2012 was found to be in a stable state in terms of pesticide use and per capita gross agricultural production values there was a "strong decoupling" state for three years, which indicated that during this period, the use of pesticides was declining with the economic growth; that is, the increase in the agricultural economy during this period did not result in much agricultural pollution. In 2013, although the use of pesticides was still declining, there was a “declining decoupling” state due to the negative growth in per capita agricultural production in Chengdu in this year. After 2013, the per capita gross agricultural production in Chengdu was annually increasing. The use of pesticides rebounded in 2016, but for the rest of the year was still decreasing; therefore, the use of pesticides and the per capita gross agricultural production values in 2014, 2015 and 2017 had good "strong decoupling" and only one "expanding negative decoupling" in 2016.

The fertilizer application and per capita gross agricultural production decoupling index was fluctuating with no obvious upward or downward trends. From 2009 to 2012, as the agricultural chemical fertilizer use declined gradually and the per capita gross agricultural production rose, the decoupling index had a short downward trend. In the last four years, two years had an ideal "strong decoupling" state. From 2012 to 2015, between 15 and 16 tonnes of agricultural chemical fertilizer was used and the annual decline was not significant. The per capita gross agricultural production value began to rise after the decline in 2013, resulting in an ideal "strong decoupling" state in 2014 and 2015. In 2016, the agricultural chemical fertilizer use increased sharply to more than 18 tonnes, which was significantly larger than the growth in the per capita gross agricultural production value; therefore, there was " expanding negative decoupling ".

The agricultural plastic film use and per capita gross agricultural production were also found to have a fluctuating trend, most of which indicated that there was a "weak decoupling" situation, with only one year indicating a "strong decoupling" ideal state, and the remaining four years being unsatisfactory. This was because except for 2017, agricultural plastic film use was increasing. In 2010 and 2014, the growth in agricultural plastic film use exceeded the growth in per capita gross agricultural production; therefore, overall, there were two years of " expanding negative decoupling " with the other years showing "weak decoupling" states.

3.3. Result Analysis

The pesticide use and per capita gross agricultural production decoupling index was relatively optimistic, with six "strong decoupling" years as the pesticide use was decreasing and the per capita gross agricultural production was increasing in most years, which indirectly indicated that the growth in the agricultural economy had little effect on agricultural pollution. The fertilizer use rate had a parabolic change from 2008 to 2017, a sharp rise (2016), and a sharp drop (2012), which resulted in an overall fluctuation in the decoupling index. However, because of the annual increases in the economy, the chemical fertilizer use had a short period of "strong decoupling" after each sharp rise or fall. The increasing use of agricultural plastic film has aggravated agricultural pollution. However, due to the reduction of the other two indicators, the relationship between agricultural pollution and agricultural economy shows a healthy and coordinated state. Agricultural pollution has not increased as a result of the growth of the agricultural economy. (line 219-272 under revised manuscript).

This is not a contextualization, thus justifying the need for this study…

Modification explanation: In the conclusion, we highlight the innovation and research significance of this paper, and further discuss the practical significance based on the previous conclusions. The contents of the conclusion correspond to the results of the research in the abstract, and also correspond to the significance of the research in the introduction.

This paper is devoted to exploring the relationship between agricultural economic development and agricultural pollution, and discussing whether it endangers public health. Therefore, the main innovations of this paper are as follows. The relationship between economic growth and ecological pollution was studied from an agricultural perspective in Chengdu, a core agricultural city in Western China, using a decoupling index to explore the key factors affecting the decoupling effect and to identify the direct relationships between the agricultural economy and agricultural pollution. A new model for measuring agricultural economy and agricultural pollution is proposed and validated with the data from 2008 to 2017. The following conclusions were drawn from the analyses.

The agricultural economy in Chengdu, Sichuan Province, China was found to be developing each year and the agricultural pollution indicators had variable changes. The use of pesticides was declining as shown by the decoupling index for pesticide use and per capita gross agricultural production calculation of a "strong decoupling" ideal state. The decoupling index for agricultural chemical fertilizer use and per capita agricultural production was also in a relatively ideal state. Although the chemical fertilizer use showed a parabolic changing state over the ten years, after each sharp rise and fall, it was in a gentle decline, which meant that there were "strong decoupling " states in some years. However, in recent years, the use of plastic film has shown a bad trend. Because of the increase in plastic film use, there were some "strong negative decoupling" years and some "expanding negative decoupling" years; however, around 50% of the time, the growth rate in plastic film use did not exceed the growth rate in the per capita gross agricultural production, which was why there were "weak decoupling" states for half the time. Generally speaking, the agricultural pollution in Chengdu was declining slowly, which coupled with the annual increase in the agricultural economy, indicated generally healthy and ideal trends. Therefore, the factor changes indicated that the main pollution indicators and the agricultural economic indicators were showing a relatively ideal trend. This shows that the use of pesticides and fertilizers in agricultural development in Chengdu has been well controlled. Although agricultural plastic film usage had an upward trend, the agricultural economic indicator growth was faster than the agricultural pollution indicator growth, which was an ideal decoupling index relationship. The use of pesticides and fertilizers has increased agricultural productivity, while it has a negative impact on public health. Pesticides and fertilizers can cause water, soil and air pollution, add heavy metals to the soil and may further cause food security problems. The situation of agricultural pollution in Chengdu has not worsened with the development of agricultural economy. Effective control of agricultural pollution in Chengdu will have a positive impact on public health. (line 274-309 under revised manuscript).

See that expressions as “was relatively poor.” Must be revised. Where is clarity.

Modification explanation: We rewrote this sentence and made the meaning clearer.

However, in recent years, the use of plastic film has shown a bad trend. (line 289 under revised manuscript).

See that all this must change…

Modification explanation: We have added the necessary content to this section.

Author Contributions: Conceptualization, S.L., Q.G. and S.Y.; methodology, S.L. and Q.G; validation, S.L. and Q.G; formal analysis, S.L. and Q.G; data curation, S.L.; writing—original draft preparation, S.L.; writing—review and editing, S.L., Q.G. and S.Y.; supervision, Q.G. and S.Y.

Funding: This research was funded by National Social Science Foundation of China (15XJY011).

Conflicts of Interest: The authors declare no conflicts of interest. (line 310-317 under revised manuscript).

I ask authors to consider looking at the text from an independent perspective, analysing all aspects in a global manner, particularly in connecting the subject to the scope of this journal, called… “Int. J. Environ. Res. Public Health”. Heath is not really discussed, rather sustainability…

Check the end…

“The relationship between the regional agricultural economic development and sustainable 275 regional agricultural development needs to be explored to determine the impact of the changes in the 276 regional agricultural economy and agricultural pollution on regional education. The influencing 277 factors for the changes in the agricultural pollution indicators also need to be further explored.”

Modification explanation: Your suggestion is very valuable. We revised the abstract, introduction, result analysis and conclusion from the perspective of public health. We have revised the paragraph you said.

The development of agricultural economy has not caused much damage to the agricultural environment. In addition, the reduction of agricultural pollution has also played a good role in public health. (line 25-27 under revised manuscript).

Modern agriculture depends heavily on pesticide use and chemical fertilizer, which have successfully increased roductivity but led to increasing concerns regarding the environment and human health [6]. Therefore, it is necessary to study the relationship between economic growth and environmental pollution as the destruction of the ecological environment can restrict economic development. Chengdu is a core city in the southwest of China. It has a good level of agricultural development and a high agricultural status in the western region. However, there is no analysis of the relationship between the agricultural economy and the agricultural environment in Chengdu in the previous literature. In order to better measure the relationship between agricultural economy and agricultural pollution, realize the sustainable development of the agricultural economy and explore public health issues, this paper use decoupling index [7] to examine this relationships and provide a new model for the study of economic growth and environmental pollution. (line 44-55 under revised manuscript).

Pesticide use contributes to increased agricultural production and improved product quality, but overuse or abuse of pesticides has been related to negative externalities such as human health effects and external ecological effects [6]. Heavy metals may be added to soils in agricultural fertilizers and pesticides, soil amendments, or organic fertilizers [37].

The use of pesticides and fertilizers has increased agricultural productivity, while it has a negative impact on public health. Pesticides and fertilizers can cause water, soil and air pollution, add heavy metals to the soil and may further cause food security problems. The situation of agricultural pollution in Chengdu has not worsened with the development of agricultural economy. Effective control of agricultural pollution in Chengdu will have a positive impact on public health. (line 300-305 under revised manuscript).

The relationship between the agricultural economic development and public health needs to be explored to determine the impact of the changes in the regional agricultural economy and agricultural pollution on resident health and food security. The influencing factors for the changes in the agricultural pollution indicators also need to be further explored. (line 306-309 under revised manuscript).

Other Modifications:

We have adjusted many sentence structures to make the article more fluent. We revised the title to make it more accurate.

We tried our best to improve the manuscript and made some changes in the manuscript. These changes will not influence the content and framework of the paper.

We appreciate for Editors/Reviewers’ warm work earnestly, and hope that the correction will meet with approval.

Once again, thank you very much for your comments and suggestions.

Sipan Li

Sichuan University

2019-10-4
